



# Impact of desert dust on new particle formation events and cloud condensation nuclei budget in dust-influenced areas

Juan Andrés Casquero-Vera[1,2,3], Daniel Pérez-Ramírez[2,4], Hassan Lyamani[2,4], Fernando Rejano[2,4], Andrea Casans[2,4], Gloria Titos[2,4], Francisco José Olmo[2,4], Lubna Dada[1,5], Simo Hakala[1], Tareq Hussein[1,6], Katrianne Lehtipalo[1], Pauli Paasonen[1], Antti Hyvärinen[7], Noemí Pérez[3], Xavier Querol[3], Sergio Rodríguez[8,9], Nikos Kalivitis[10], Yenny González[11,8], Mansour A. Alghamdi[12], Veli-Matti Kerminen[1], Andrés Alastuey[3], Tuukka Petäjä[1], and Lucas Alados-Arboledas[2,4]

[1]Institute for Atmospheric and Earth System Research (INAR)/Physics, University of Helsinki, Helsinki, Finland
[2]Andalusian Institute for Earth System Research (IISTA-CEAMA), University of Granada, Granada, Spain
[3]Institute of Environmental Assessment and Water Research (IDAEA), CSIC, Barcelona, Spain
[4]Department of Applied Physics, University of Granada, Granada, Spain
[5]Laboratory of Atmospheric Chemistry, Paul Scherrer Institute, Villigen, Switzerland
[6]Department of Physics, The University of Jordan, Amman, Jordan
[7]Atmospheric Composition Research, Finnish Meteorological Institute, Helsinki, Finland
[8]Izaña Atmospheric Research Centre, Agencia Estatal de Meteorología, Santa Cruz de Tenerife, Spain
[9]Group of Atmosphere, Aerosols and Climate-AAC, IPNA CSIC, Tenerife, Spain
[10]Environmental Chemical Processes Laboratory, University of Crete, Heraklion, Greece
[11]CIMEL Electronique, Paris, 75011, France
[12]Department of Environmental Sciences, Faculty of Meteorology, Environment and Arid Land Agriculture, King Abdulaziz University, Jeddah, Saudi Arabia

*Correspondence to*: Juan Andrés Casquero Vera (juan.casquero@helsinki.fi; casquero@ugr.es)

**Abstract.**

Detailed knowledge on the formation of new aerosol particles in the atmosphere from precursor gases, and their subsequent growth, commonly known as new particle formation (NPF) events, is one of the largest challenges in atmospheric aerosol science. High pre-existing particle loadings are expected to suppress the formation of new atmospheric aerosol particles due to high coagulation and condensation (CS) sinks. However, NPF events are regularly observed in conditions with high concentrations of pre-existing particles and even during intense desert dust intrusions that imply discrepancies between the observations and theory. In this study, we present a multi-site analysis of the occurrence of NPF events under the presence of desert dust particles in dust-influenced areas. Characterization of NPF events at 5 different locations highly influenced by desert dust outbreaks was made under dusty and non-dusty conditions by using continuous measurements of aerosol size distribution in both fine and coarse size fractions. Contrary to the common thought, our results show that the occurrence of NPF events is highly frequent during desert dust outbreaks, showing that NPF event frequencies during dusty conditions are similar to those observed during non-dusty conditions. Furthermore, our results show that NPF events also occur during intense desert dust outbreaks at all the studied sites, even at remote sites where the amount of precursor vapours is expected to be low. Our results show that the condensation



sink associated with coarse particles ($CS_C$) represents up to the 60% of the total CS during dusty conditions, which highlights the importance of considering coarse fraction particles for NPF studies in desert dust influenced areas. However, we did not find a clear pattern of the effect of desert dust outbreaks on the strength of NPF events, with differences from site to site. The particle growth rate (GR) did not present a clear dependence on the CS during dusty and non-dusty conditions. This result, together with

the fact that desert dust has different effects on the growth and formation rates at each site, suggest different formation and growth mechanisms at each site between dusty and non-dusty conditions, probably due to differences in precursor vapours origins and concentrations as well as changes in the oxidative capacity of pre-existing particles and their effectiveness acting as CS. Further investigation based on multiplatform measurement campaigns and chamber experiments with state-of-the-art gaseous and particulate physical and chemical properties measurements is needed to better understand the role of catalyst components present

in desert dust particles in the process of NPF. Finally, our results suggest that the contribution of NPF events to cloud condensation nuclei (CCN) budget is larger during dusty conditions than during non-dusty conditions. Therefore, since desert dust contributes to a major fraction of the global aerosol mass load, and since there is a foreseeable increase of the frequency, duration, and intensity of desert dust episodes due to climate change, it is imperative to improve our understanding on the effect of desert dust outbreaks on NPF and CCN budget for better climate change prediction.

## 1 Introduction

Atmospheric new particle formation (NPF) involves the production of molecular clusters from condensable vapours, such as sulphuric acid, ammonia, amines, volatile organic compounds and other trace gases capable of forming low-volatility complexities, and subsequent growth of these clusters to larger sizes (Kulmala et al., 2013; Kirkby et al., 2016). Ultimately, the newly formed particles can grow into the cloud condensation (CCN) size range by coagulation and condensation of

additional vapours (Riipinen et al., 2011), affecting the climate via aerosol-cloud interactions (Kerminen et al., 2018; Bellouin et al. 2020; Rejano et al., 2021). Despite the advancements in theoretical knowledge of NPF steps described by Kulmala et al. (2014), large discrepancies have been found between the expected and observed properties of NPF under different atmospheric conditions (Nieminen et al., 2018). Despite the advancements in theoretical knowledge of NPF steps described by Kulmala et al. (2014), large discrepancies have been found between the expected and observed properties of NPF under different

atmospheric conditions (Nieminen et al., 2018).

Theoretically, high pre-existing aerosol particle concentrations are expected to inhibit NPF, by enhancing the scavenging of both condensable vapours and small clusters formed from these vapours (Kulmala et al., 2017). The survival probability of growing clusters and nanoparticles depends on the ratio between their growth rate (GR; how fast the particles grow to larger sizes) and their loss rate due to scavenging by pre-existing particles, described by the condensation sink (CS; amount of surface area of pre-

existing particles). Therefore, high CS might influence NPF by inhibiting this process (e.g., Du et al., 2022; Tuovinen et al., 2020). However, NPF events are regularly observed under conditions with high concentrations of pre-existing particles, such as in highly





polluted megacities or during strong desert dust outbreaks (e.g., Kulmala et al., 2017; Nie et al., 2014; Casquero-Vera et al., 2020). A possible explanation could be that the scavenging of either vapours responsible for the cluster formation and initial growth or growing cluster by available pre-existing particles is less efficient than estimated (Kulmala et al., 2017; Gani et al., 2020). In general, the loss of a certain vapour could be affected by the characteristics of the pre-existing particles. However, when the CS

is calculated, traditionally, only the particle number size distribution (PNSD) is considered. This assumes that other factors, such as the particles' morphology, physical state and chemical composition, do not affect the CS.

Among aerosol sources, desert dust particles contribute to a major fraction of the global coarse mode aerosol load, with an emission rate into the atmosphere estimated at 5000 Tg/yr (Kok et al., 2021). North African and Arabian areas are among the most important sources of dust in the world. Due to their proximity, the Mediterranean basin, North Atlantic and Arabian regions

are frequently affected by dust transport from these desert regions, and it is foreseeable that the frequency, duration, and intensity of these desert dust episodes will increase due to climate change (Salvador et al., 2022). Although NPF is not expected to occur during desert dust outbreaks (because of high coagulation and condensation sink by dust; Zamora and Kahn, 2020), NPF events have been observed in remote sites under dusty conditions (e.g., Nie et al., 2014; Baalbaki et al., 2021; Casquero-Vera et al., 2020), implying discrepancies between the observations and theoretical expectations.

The climatic effects of desert dust and atmospheric NPF have been thought to be disconnected from each other, however, high dust loadings can affect NPF in opposing ways. Desert dust can suppress photochemical processes by scavenging reactive gases and condensable vapours and, hence, can inhibit NPF occurrence (de Reus et al., 2000; Ndour et al., 2009). But also, the presence of $TiO_2$ and $Fe_2O_3$ (which are common components of desert dust) under UV light could promote the occurrence of NPF by acting as a site for heterogeneous photochemistry promoting the formation of OH radicals, which initiate the

conversion of $SO_2$ to $H_2SO_4$ (Dupart et al., 2012; Nie et al., 2014). A clear association between high dust loading and the occurrence or strength of NPF has not yet been established.

Furthermore, NPF events are an important mechanism producing CCN (source of more than the 50% of CCN at low supersaturations; Gordon et al., 2017) and desert dust particles are efficient CCN as well (Levin et al., 2005). In this sense, desert dust transport can affect the CCN budget via 1) depleting the reservoir of vapours required to form CCNs from particles

originating from atmospheric NPF, and via 2) coating of desert dust particles by soluble material which enhances the CCN activity of dust particles themselves (Nenes et al., 2014 and references therein). Thus, since NPF is a major source of CCN and desert dust is a major component of the global aerosol load, understanding the interplay between desert dust and NPF events and their impact on CCN concentrations and climate is of special relevance.

The objective of this work is to study the occurrence of NPF events under the presence of desert dust particles in different

dust-influenced areas. For this purpose, we characterize NPF events under dusty and non-dusty conditions at 5 different locations highly influenced by desert dust. The observatories used in this study have continuous measurements of aerosol size distribution in both fine and coarse fractions (from 10 nm to 10 µm). We identify and characterize NPF events at each site and



analyse and discuss the influence of desert dust outbreaks on the occurrence frequency, strength and characteristics of NPF events at these desert dust-influenced areas. Finally, the influence of desert dust outbreaks associated with NPF events on the potential CCN budget will be analysed. This is the first study of the influence of desert dust on CS and its effectiveness on the inhibition/promotion of NPF events in multiple dust-influenced areas.

## 2 Measurements and methods

### 2.1 Measurement sites and instrumentation

In the present study, particle number size distribution (PNSD) data in the size range of 10 nm - 10 µm from 5 sites distributed in the Mediterranean basin, North Atlantic and Arabian regions (Fig. 1) were analysed. All the sites operated similar instrumentation and the observations followed guidelines set by ACTRIS (Aerosols, Clouds and Trace gases Research Infrastructure; https://www.actris.eu/) for in situ aerosol number size distribution measurements (Wiedensohler et al., 2012). The observation sites (Fig. 1) and instruments are shortly described below.

- The Izaña observatory (IZO station; 28.30º N, 16.49º W) is located on the island of Tenerife (Spain), at 2373 m a.s.l. The Izaña station is part of the GAW (Global Atmospheric Watch, World Meteorological Organization) network and has accumulated approximately 15 years of measurements of aerosol properties, trace gases and radiation. The observatory remains almost permanently above the marine stratocumulus layer typical of the subtropical oceans, and it is frequently affected by severe desert dust outbreaks due to its proximity to the African continent. NPF events has been previously observed with an annual event frequency of 30% (García et al., 2014). The analysed particle number size distribution dataset collected at IZO covers 366 days from 26-Jun-2014 to 26-Jun-2015. PNSD within the range 10–500 nm was measured with a Scanning Mobility Particle Sizer (SMPS, TSI model 3996) and with an Aerodynamic Particle Sizer (APS, TSI model 3321) for the range 0.5–10 µm.

- The Sierra Nevada site (SNS station; 37.10º N, 3.39º W, 2500 m a.s.l.) is located on the northern slope of the Sierra Nevada mountain range, ∼ 5 km north-west of the Veleta summit (3398 m a.s.l.) and ∼ 20 km south-east of the city of Granada (Spain). This station is part of the AGORA (Andalusian Global ObseRvatory of the Atmosphere) observatory and is representative of the south-western European free troposphere. The site is frequently affected by local aerosol particles transported from lower altitudes and NPF events, especially in summer and at midday (Casquero-Vera et al., 2020). Additionally, natural dust outbreaks from North Africa are very common in the area (e.g., Lyamani et al., 2010; Benavent-Oltra et al., 2021; Cazorla et al., 2017). The analysed PNSD dataset collected at SNS covers 90 days from 08-Jun-2017 to 05-Sep-2017. PNSD within the range 10–500 nm was measured with a SMPS (TSI model 3938) and with an APS (TSI model 3321) for the range 0.5–10 µm.

- The Montseny site (MSY station; 41.77º N, 2.36º E, 720 m a.s.l.) is an ACTRIS regional background station located in the mountainous area of Montseny, within a regional natural park about 50 km N-NE of the city of Barcelona (Spain) and 25 km from the Mediterranean coast. The site represents typical regional background aerosol conditions of the Western





Mediterranean Basin characterized by severe pollution episodes affecting not only the coastal sites closest to the emission sources (e.g., Pandolfi et al., 2011) and desert dust intrusions (Ealo et al., 2016; Titos et al., 2017). The analysed PNSD dataset collected at MSY covers 365 days from 01-Jan-2017 to 31-Dec-2017. PNSD within the range 10–900 nm was measured with a Mobility Particle Size Spectrometer (MPSS; TROPOS) and with an Optical Particle Sizer (OPS; GRIMM 5 180) for the range 0.9–20 µm.

-   The Amman site (AMM station; 32.01º N, 35.87º E, 1000 m a.s.l.) is an urban background station located in the Environmental and Atmospheric Research Laboratory (EARL), in the north part of the city of Amman (Jordan). The site represents urban background conditions, affected by local anthropogenic sources and, frequently, influenced by long-range transport of natural dust from North Africa, the Arabian Peninsula, and the Levant (Hussein et al., 2020a). The analysed PNSD dataset collected 10 at AMM covers 365 days from 01-Aug-2016 to 31-Jul-2017. PNSD within the range 10-400 nm was measured with a SMPS (TSI model 3910) and with an OPS (TSI model 3330) for the range 0.4-10 µm.

-   The Hada Al Sham site (HAS station; 21.80º N, 39.73º E; 254 m a.s.l.) is a rural background site located in the Agricultural Research Station of King Abdulaziz University, in the small town of Hada Al Sham (Saudi Arabia). Sparsely inhabited desert-like areas cover the inland in the N–SE direction from the measurement site, but the coastal regions in the western 15 sector are densely populated (Lihavainen et al., 2016). There are no major sources of anthropogenic emissions in the immediate vicinity and the frequency of occurrence of NPF events in the area is one the highest in the world (Hakala et al., 2019). The analysed PNSD dataset collected at HAS covers 327 days from 27-Jun-2013 to 31-May-2014. PNSD within the range 10-900 nm was measured with a twin DMPS (Differential Mobility Particle Sizer; Aalto et al., 2001) and with an APS (TSI model 3321) for the range 0.9-10 µm.

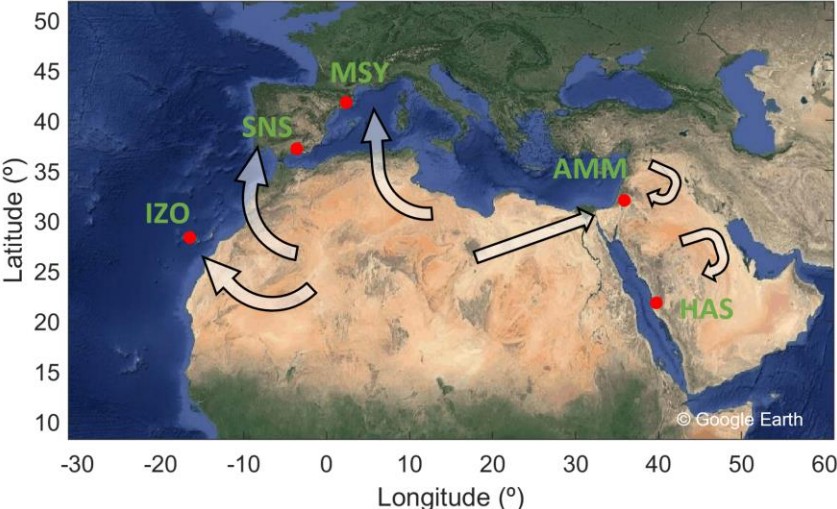

**Figure 1. Location of measurement sites from western to eastern: Izaña (IZO), Sierra Nevada (SNS), Montseny (MSY), Amman (AMM) and Hada Al Sham (HAS). Arrows indicate main desert dust source areas and transport ways for each measurement site based on Griffin (2007).**



## 2.2 Data analysis

### 2.2.1 NPF classification and characterization

The classification of NPF event days was done by visual inspection of the daily particle number size distribution data according to the guidelines presented by Dal Maso et al. (2005). According to this classification criteria, days are classified into four

groups: event (E), non-event (NE), undefined (UN) and bad-data days (BD). (1) "E" days are days during which sub-25 nm particle formation and their consequent growth are observed, (2) "NE" days are days on which neither new growing modes nor production of sub-25 nm particles are observed, (3) "UN" days are the days which do not fit either of the previous classes, and (4) "BD" days are the days during which data are not valid or inexistent. In addition, event days are separated into two different groups: class I and II events. NPF events are classified as class I events when the NPF growth rate retrieval is possible

and as class II when it is not possible.

The automatic DO-FIT algorithm (Hussein et al., 2005) was used to describe the measured particle number size distributions by fitting multiple log-normal distributions to the measured data. The geometric mean diameters of the fitted distributions were then used for calculating particle growth rates during NPF events. This was done by calculating the slope of the linear fit to the geometric mean diameters as a function of time, which were identified to represent the growing particle mode formed in

an NPF event. Thus, the growth rate (GR) was obtained as:

$$GR_{\Delta D_p} = \frac{dD_p}{dt} = \frac{\Delta D_p}{\Delta t} \qquad\qquad\qquad \text{Eq. (1)}$$

where $D_p$ is the representative diameter of the NPF mode at time $t$. In this work, the GR in the 10-25 nm range (GR$_{10-25}$ named as GR from now on for simplicity) was calculated. The uncertainty in the calculated GR was estimated to be 8% for the 7-20 nm size ranges (Yli-Juuti et al., 2011).

The formation rate ($J_{D_p}$) is defined as the flux of particles past the lower limit of the size range ($\Delta D_p$), and it is obtained by adding up the observed change in the observed particle number concentration and the losses of particles due to coagulation and growth out of the size range ($\Delta D_p$) and it is calculated following the methodology described by Kulmala et al. (2012):

$$J_{D_p} = \frac{dN_{D_p}}{dt} + CoagS_{D_p} \cdot N_{D_p} + \frac{GR}{\Delta D_P} \cdot N_{D_p} \qquad\qquad\qquad \text{Eq. (2)}$$

where the first term on the right hand side represents the rate of change of particle concentration with time (where $N_{\Delta D_p}$ is the

particle number concentration in the size range $\Delta D_p$); the second term describes the loss of particles due to coagulation with larger aerosol particles (where $CoagS_{\Delta D_p}$ is the coagulation sink); and the third term considers the growth out of the considered size range. In this study, we calculated the formation rate at diameter ($D_p$) 10 nm ($J_{10}$ named as $J$ from now on for simplicity).



Finally, the condensation sink (CS) describes how rapidly vapour molecules may condense onto pre-existing aerosols. CS is dependent on the effective surface area of the pre-existing particle size distribution (Kulmala et al., 2012). Accordingly, the CS was calculated from each size distribution as:

$$CS = 2\pi D \int_{D_{min}}^{D_{max}} D_p \; \beta_M \; N_{D_p} \; dD_p = 2\pi D \sum_{D_p} D_p \; \beta_M \; N_{D_p} \qquad \text{Eq. (3)}$$

where $D$ is the diffusion coefficient of condensable vapour, that is assumed to be sulfuric acid, and $\beta_M$ is the transitional correction factor (Fuchs and Sutugin, 1971) which is dependent on the mean free path of vapour molecules and aerosol diameter (Kulmala et al., 2001; Pirjola et al., 1999). In order to study the impact of desert dust on CS, we calculated the CS for fine ($CS_F$ for diameters <1 µm) and coarse ($CS_C$ for diameters >1 µm) particles.

**2.2.2 Desert dust analysis**

The presence of desert dust particles was determined by applying a procedure similar to the standard one used by the European Commission to identify the natural episodes of desert dust in order to perform the discounts in the calculation of the annual average of $PM_{10}$ (Air Quality Directive 2008/50/EC; Viana et al., 2010; Escudero et al., 2007; Querol et al., 2009 and references therein). First, for detecting desert dust intrusions over a specific area, satellite images such as MODIS (https://modis.gsfc.nasa.gov/data/dataprod/mod04.php) and aerosol index of OMI (Ozone Monitoring Instrument;

ftp://toms.gsfc.nasa.gov/pub/omi/images/aerosol/), aerosol forecast models like BSC-DREAM (https://ess.bsc.es/bsc-dust-daily-forecast) and NAAPs (https://www.nrlmry.navy.mil/), as well as synoptic meteorological charts are consulted. If the presence of dust at surface level (surface concentrations >5 µg m⁻³) is shown by at least one model, this specific day is classified as possible day with influence of desert dust (e.g., Viana et al., 2010). It is important to note that in this study we only analysed desert dust events that reached the ground level at the different sites and thus desert dust intrusions at high altitude level are

not included in this study. To this end, ground level measurements of APS/OPS are used. In this sense, the presence of dust is confirmed at ground level if an increase of the coarse mode particle concentration and of the ratio between the coarse and fine mode particle number concentrations is observed. Finally, to determine the source regions of the detected desert dust, 120 h back-trajectories were calculated daily for the detected dusty days with the HYSPLIT model (Stein et al., 2015). This is a qualitative method to detect the presence of dust, but it is not possible to quantify the intensity of the outbreak. However, in

order to identify the impact of severe desert dust outbreaks on the occurrence of NPF events, we further classified some of the days as intense dusty conditions. In this sense, there is no standard definition or threshold in the literature for dust and intense dust episodes. Thus, in this study we classified intense dusty days as those dusty days with a condensation sink of coarse mode particles ($CS_C$) above the 75[th] percentile at each site.



# 3 Results and discussion

## 3.1 Frequency and seasonality of NPF events

Table 1 presents the number and frequency of event, non-event and undefined NPF days at the 5 studied sites. The annual NPF event frequency ranged from 34% to 68%, being the lowest at IZO and AMM which are sites having very different

characteristics. IZO is a high-altitude remote station located in Tenerife Island with very minor emission sources close to the station. Aerosol particles and vapours observed at IZO station are mainly emitted at lower altitudes by natural and anthropogenic sources and then transported to the station by orographic thermal-buoyant upward flows during daytime (García et al., 2014). By contrast, AMM is an urban background station located in the city of Amman (Jordan) that is mainly affected by local anthropogenic emissions but also by long-range transport episodes (Hussein et al., 2020a). Thus, although the NPF

event frequency observed at IZO and AMM sites are similarly low in comparison to the other studied sites, the reasons for the low frequency of NPF events at both sites are expected to be completely different. In this sense, the low frequency of NPF events at IZO is associated with low concentrations of precursor gases at this remote high-altitude site (e.g., García et al., 2014), while the low frequency of NPF at the AMM site is due to high concentrations of pre-existing particles associated with local anthropogenic emissions at this urban site (Hussein et al., 2020b). It is worth mentioning that the observed values for the

study period at these two sites are in agreement with those reported by García et al. (2014) on a climatological study of NPF events at IZO and by Hussein et al. (2020b) at AMM.

**Table 1. Number (*n*) and frequency of occurrence (*f*) of event, non-event or undefined NPF days and bad-data days observed at the 5 studied sites.**

|  | IZO | | SNS | | MSY | | AMM | | HAS | |
|---|---|---|---|---|---|---|---|---|---|---|
|  | *n* | *f* | *n* | *f* | *n* | *f* | *n* | *f* | *n* | *f* |
| **Event** | 109 | 34% | 54 | 69% | 195 | 62% | 110 | 34% | 197 | 68% |
| Class I | 43 | 13% | 37 | 47% | 46 | 15% | 110 | 34% | 190 | 66% |
| Class II | 66 | 21% | 17 | 22% | 149 | 47% | - | - | 7 | 2% |
| **Non-event** | 132 | 41% | 20 | 26% | 62 | 20% | 121 | 38% | 26 | 9% |
| **Undefined** | 79 | 25% | 4 | 5% | 58 | 18% | 91 | 28% | 67 | 23% |
| **Bad-data** | 46 | - | 12 | - | 50 | - | 43 | - | 37 | - |

The highest annual NPF event frequency (68%) was observed at HAS, twice that at AMM (the other studied site located in Middle

East area). This is an unexpected result because HAS is a rural background station located in an arid area characterized by high aerosol particle loads and with no major sources of anthropogenic emissions in the immediate vicinity (Hakala et al., 2019), whereas AMM is an urban background site with relatively high abundances of precursor vapours due to local urban anthropogenic



emissions (Hussein et al., 2020b). The high NPF event frequency observed in HAS site is in good agreement with that reported by Hakala et al. (2019), who concluded that the NPF event frequency at HAS is among the highest event frequencies in the world. Such high NPF event frequency at HAS was previously explained by the typically prevailing conditions of clear skies and high solar radiation, in combination with sufficient amounts of anthropogenic precursor vapours transported from the coastal urban

and industrial areas (~60 km away) (Hakala et al., 2019). In this sense, although HAS presents high $PM_{10}$ concentrations (annual mean of 95 µg/m$^3$; Lihavainen et al., 2016), the high event frequency observed at this site (68%) indicates that the concentration and chemical composition of pre-existing particles does not limit the occurrence of NPF events.

The frequency of NPF events at the HAS site did not show a clear seasonal pattern, besides being relatively high throughout the year (53-77%; Fig. S1 in the Supplement). This implies that the NPF events are not limited by any factor with a strong

seasonal variability and that probably the origin of precursor vapours (and NPF events) is mainly from anthropogenic sources with no seasonal variability. By contrast, the NPF frequencies at the rest of sites show clear seasonal patterns, with NPF events occurring mainly during late spring and early summer. This seasonal variability is in agreement with the global review of NPF events by Nieminen et al. (2018), who reported the highest NPF occurrence during spring/summer and lowest during autumn/winter. Several previous studies related this seasonal variability on NPF events frequency with the seasonal variability

on solar radiation and emissions of precursor gases from biogenic sources (e.g., Dada et al., 2017; Jokinen et al., 2022).

The high NPF event frequency observed at SNS during summer (69%) is associated with the high intensity of solar radiation and the increase of precursor gases transport from lower altitudes during summer season (Casquero-Vera et al., 2020; 2021). It is important to remark that comparing the summer (June, July and August) event frequencies of the western sites (SNS, MSY and IZO), the summer event frequency of MSY was the highest among these sites (80%), followed by SNS (69%) and

IZO (40%). In this sense, the lowest summer event frequency observed at IZO station (half of the summer event frequency observed at MSY) suggests larger sources of anthropogenic and biogenic vapours affecting MSY and SNS and/or that other mechanisms could be responsible for the formation of new particles at IZO (e.g., formation promoted by marine emissions).

In addition to the aforementioned local differences in the emissions of precursor vapours and the atmospheric conditions that could favour or inhibit the formation of new particles and explain the differences in the NPF occurrence frequency between

the studied sites, long-range air masses transport has been identified to play different roles on the occurrence of NPF events depending on the study area. For example, Petäjä et al. (2022) and Hakala et al. (2022) showed that the occurrence of NPF events over a boreal forest and Beijing megacity, respectively, are limited to cleaner air mass transport and that highly loaded air masses inhibit the occurrence of NPF events. These studies agree with the common thought that high loading of pre-existing particles and NPF do not usually co-exist, since high concentrations of particles tend to scavenge both low-volatile vapours

and small molecular clusters efficiently (e.g., Nie et al., 2014). Several existing observations confirm this expectation, since at most measurement sites the average value of CS was found to be lower on NPF event days compared with non-event days (Kerminen et al., 2018 and references therein). However, as can be seen below, NPF events were also observed during long-



range transport of desert dust particles over the different studied sites; episodes that are usually characacterized by relativelly high aerosol loads. Thus, in the following in depth analysis of the role of desert dust intrusions in promoting or inhibiting the occurrence of NPF events will be done.

## 3.2 Impact of desert dust outbreaks on NPF event frequency

The dust emitted in the dust-belt (Prospero et al., 2002; Dominguez-Rodriguez et al., 2020) frequently expands over the Mediterranean basin, North Atlantic and Arabs regions. Thus, these areas, where the studied sites are located, are some of the areas most influenced by desert dust in the world (Dominguez-Rodriguez et al., 2020), which may affect the frequency of NPF events. In fact, the results show that the studied sites were affected by dust outbreaks on 27% (total number of days classified as dusty: 87 days), 83% (64 days), 23% (72 days), 68% (218 days) and 89% (258 days) of the analysed days at IZO, SNS, MSY,

AMM and HAS, respectively. The annual and seasonal dust intrusion frequencies (Fig. S2) over western sites (IZO, SNS and MSY) are in good agreement with those previously reported for these areas for a 15-year period by Querol et al. (2019). Although there are no previous long-term studies reporting the frequency of occurrence of desert dust outbreaks at HAS and AMM sites, Hussein et al. (2014) and Hussein et al. (2018) reported highest coarse particle concentrations and/or $PM_{10}$ concentrations during spring and autumn at AMM and for the Feb-May period at Jeddah city (~60 km away from HAS site), respectively. These results

agree with the periods when we observe the highest dust outbreaks frequencies at AMM and HAS (Fig. S2).

To analyse the potential impact of desert dust intrusions on the occurrence of NPF events, Figure 2 shows the NPF event frequency for the whole measurement period and the percentage of NPF event days observed during non-dusty and dusty conditions at each measurement site. In addition, in order to evaluate the possible dependence of the occurrence of NPF on the intensity of dust intrusion, and to investigate whether the most intense desert dust episodes can inhibit the occurrence of NPF

events, we included in our analysis the NPF events that occurred during the most intense desert dust periods at each site. A dusty day at each specific site was classified as intense dusty day when the $CS_C$ in this day was above the 75$^{th}$ percentile of the dusty days at this specific site. According to this criterion, 16, 14, 16, 49 and 59 days at IZO, SNS, MSY, AMM and HAS were classified as intense dusty days.

The results show that the percentage of NPF events that occur during dusty conditions is similar to that observed during non-

dusty conditions in all the studied sites, except in SNS site (Fig. 2). Nevertheless, it is important to note that the percentage of NPF events that occur during intense dusty conditions in SNS was relatively high (79%) and almost similar to the observed during non-dusty conditions (Fig. 2). Thus, these results point to a minimal impact of desert dust outbreaks on the occurrence of NPF events at the studied sites, including SNS site. Additionally, the percentage of NPF events that occurred during intense dusty conditions at SNS, MSY and HAS was relatively high (>70%), being similar to or even higher than what was observed

during non-dusty and dusty conditions in these sites (Fig. 2). This evidences that the intensity of desert dust intrusions does not represent a determinant factor affecting the occurrence of NPF events at SNS, MSY and HAS sites. It is worth to mention





that even if we increase the $CS_C$ threshold to the 90[th] percentile for the classification of intense dusty days at these sites, the results still similar to those observed for the 75[th] percentile, showing that the occurrence of NPF events is even possible during the strongest dust episodes. This result points to the presence of sufficient amounts of precursor vapours to compete with the larger available surface of pre-existing particles or to a potential inefficiency of desert dust particles in acting as CS at these

sites. Therefore, the occurrence of NPF events at these sites is probably not only limited to highly polluted dust plumes (as suggested by Nie et al., 2014), and they can even occur in remote sites during desert dust outbreaks. By contrast, in IZO and AMM sites the percentages of NPF events observed during dusty conditions were similar to those obtained during non-dusty conditions but a significant decrease was observed during intense dusty conditions. Thus, this result evidences that the intensity of desert dust can limit the occurrence of NPF events at IZO and AMM, which could be explained by a significant reduction or

a limited amount of precursor gases to compete with the increase of available surface of pre-existing particles. In this sense, in the following we will analyse the role of desert dust in increasing the surface area of pre-existing particles at each site and it impact on NPF events.

It is worth to mention that the seasonal variation of NPF and desert dust outbreaks could be different, with desert dust outbreaks occurring during periods of low NPF event frequency and vice-versa (Fig. S1 and S2), thus affecting the NPF event frequency

at each site. However, since the seasonal variation of desert dust outbreaks at the different studied sites did not follow the same clearly marked pattern as for NPF event frequencies, we consider that there is a minimal impact of this seasonal variation. Therefore, the results show that the occurrence of mineral dust outbreaks and the presence of mineral dust do not represent a determinant factor that affect the occurrence of NPF events.

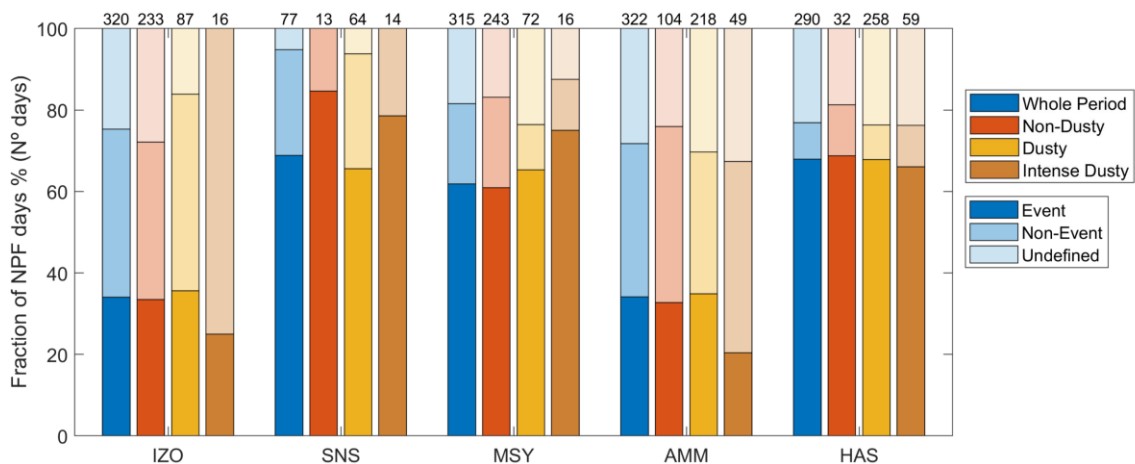

**Figure 2. Frequency of NPF event, non-event and undefined days during the whole measurement period and the percentage of NPF events during non-dusty, dusty and intense dusty conditions at the 5 studied sites. Numbers on top represent the total number of days for each category.**





### 3.3 Contribution of desert dust to CS

High pre-existing aerosol particle concentrations during desert dust outbreaks could inhibit the process of NPF by increasing the competition for available condensable gases and thus by increasing the loss rate of vapours onto pre-existing aerosol particles. This loss rate of precursor vapours is usually quantified by the condensation sink (CS; Eq. 3) that describes how rapidly vapour molecules may condense onto the surface of pre-existing aerosols. Among other factors, CS depends on the aerosol size distribution. In general, the CS is usually calculated by only considering the PNSD measured by MPSS (i.e., fine particles with $D_p$ < 800 nm). However, this can result in an underestimation of CS if the contribution of coarse particles to total ambient aerosol concentration is significant as is the case during desert dust outbreaks. For this reason, in this study we calculated the total condensation sink (CS) by using the whole measured PNSD (fine and coarse mode size ranges), and we also separated CS for fine ($CS_F$ for diameters <1 µm) and coarse ($CS_C$ for diameters >1 µm) particles.

The total CS during the whole measurement periods shows low values at the western sites with median CS values of 0.001, 0.004 (summer median) and 0.005 $s^{-1}$ at IZO, SNS and MSY, respectively. By contrast, significantly large CS values were observed at HAS and AMM with annual median values of 0.020 and 0.010 $s^{-1}$, respectively. It is important to note that although the annual median value of CS observed at IZO was the lowest among the studied sites, the frequency of NPF events at this site (34%) was also the lowest one (Table 1). Furthermore, the site with the highest median CS (HAS) had the highest NPF event frequency (68%). Thus, the results suggest that the CS alone does not explain the observed differences in the NPF frequency between the different studied sites, and that these differences are probably associated with the availability of gaseous precursors in each site, atmospheric conditions and/or with the effectiveness of pre-existing particles in scavenging gaseous precursors.

When considering only fine particles, the observed values of CS were in the range of those reported in previous studies at each measurement site (García et al., 2014; Casquero-Vera et al., 2020; Carnerero et al., 2019; Hussein et al., 2020b; Hakala et al., 2019). The overall contribution of coarse particles ($CS_C$) to the total CS during the whole analysed periods is of 12%, 10%, 2%, 2% and 17% at IZO, SNS, MSY, AMM and HAS, respectively. These results agree with previous studies at IZO and HAS, where the overall contribution of coarse particles to total CS was reported to be 8% (García et al., 2014) and 10% (only during non-event NPF days; Hakala et al., 2019). However, the impact of coarse particles on the total CS at all studied sites is expected to be significantly larger during dusty conditions. In fact, the $CS_C$ obtained during dusty conditions was on average 21, 4.5, 2.5, 1.5 and 2 times larger than that observed during non-dusty conditions at IZO, SNS, MSY, AMM and HAS, respectively. Furthermore, on average coarse particles contributed significantly to the total CS during dusty conditions at IZO and HAS sites (by 60% and 23%, respectively), while their average contributions during dusty conditions were significantly lower at SNS, MSY and AMM (13%, 3% and 2%, respectively). However, while there was significant increase of $CS_C$ during dusty conditions at all studied sites, a clear increase of the total CS during dusty conditions in comparison with non-dusty days was only observed at IZO (2 times larger during dusty than non-dusty conditions), MSY (1.5 times larger) and HAS (2 times larger). This means that the increase of $CS_C$ during dusty conditions, especially at SNS and AMM, is associated by the removal of fine particles by coagulation onto the





surface of dust particles, resulting in a reduction of CS of fine particles. In fact, $CS_F$ at these two sites (SNS and AMM) was 10% lower during dusty conditions than during non-dusty conditions. Thus, these results highlight the impact of desert dust outbreaks into the CS and the importance of considering coarse mode particles for adequate CS calculations at desert dust-influenced areas.

In order to evaluate the impact of CS on the frequency of NPF events at each station, Figure 3 shows the comparison of CS

obtained during NPF events (E) and non-NPF events (NE) for dusty and non-dusty days. The results show that, during dusty conditions, the median values of CS were 42%, 60% and 17% larger for the NPF event days than for NE days at SNS, MSY and HAS, respectively. Similarly, during non-dusty days, the median values of CS were larger at these sites during NPF event days compared to NE days, revealing that the CS was not a key factor limiting the occurrence of NPF events during dusty and non-dusty conditions at SNS, MSY and HAS. Additionally, during dusty conditions at these studied sites (SNS, MSY and HAS), the

75th and 99th percentiles of CS showed larger values on NPF events than on NE days, suggesting that the NPF events observed at these sites were not inhibited even during the strongest dusty days with highest concentrations of pre-existing particles. Thus, desert dust intrusions at SNS, MSY and HAS do not inhibit NPF but instead promote its occurrence.

As expected, the median CS observed at AMM during dusty conditions was larger for NE days (by 28%) than for NPF event days. Furthermore, at AMM site the median CS and interquartile ranges for NE days were larger than for E days during both

dusty and non-dusty conditions, suggesting that regardless of the origin of pre-existing particles (dust or anthropogenic particles), CS controls the occurrence of NPF events at this site. At the IZO site, despite during dusty conditions there were no differences in the median CS for NE and E days, the 75th and 90th percentile values of CS were larger on NE days than on E days, suggesting that highest values of CS inhibit the occurrence of NPF events at this site. This agrees with the result on event frequencies in the previous section.

Overall, our results show a dependence from site to site on the effect of pre-existing particles on the suppression/promotion of NPF events. These differences could be related to an increase/reduction of precursor vapours with the increased/reduced transport of pre-existing particles (e.g., Du et al., 2022). However, these differences could also be related to different atmospheric conditions or the chemical composition of pre-existing particles and precursor vapours, which can cause differences in the scavenging efficiency of CS and its inhibiting effect of NPF events (Tuovinen et al., 2020; Du et al., 2022). In this sense, recent results show

that indeed the CS is a key factor controlling the occurrence of NPF events, the chemical composition of the pre-existing particles affects their efficiency in removing precursor vapours (effectiveness of the CS), and thus in driving NPF events (Du et al., 2022). Nevertheless, a detailed understanding of the role of CS in NPF events remains an open question (Tuovinen et al., 2020).





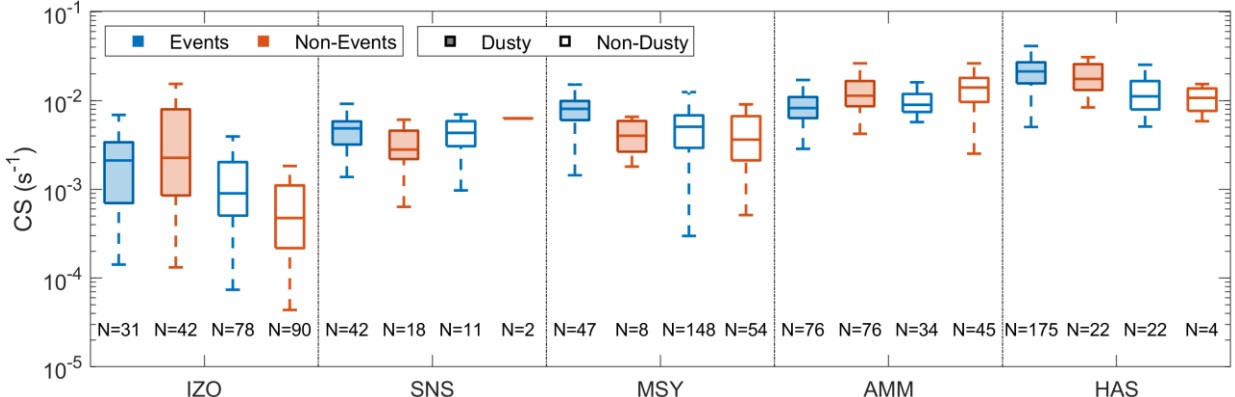

**Figure 3. Box and whisker plot of CS recorded at each measurement site during NPF events (blue) and non-events (red) and during dusty (shaded boxes) and non-dusty (non-shaded boxes) condition days. The line represents the median of the data and the lower and upper edges of the box represent the 25th and 75th percentiles of the data, respectively. The whiskers go from the end of the interquartile range to the furthest observation within 1.5× interquartile range. N represents the total number of days for each category.**

### 3.4 Impact of desert dust on particle formation and growth rates

In addition to influencing the frequency of occurrence of NPF events, desert dust could change the strength and other characteristics of the NPF events occurring during dusty conditions. The strength of a NPF event is determined by its duration, the rate at which new particles are formed ($J$) and the growth rate (GR) of these particles into larger sizes. In this sense, the increased contribution of desert dust particles during dusty conditions could imply the inhibition or enhancement of the NPF process. On the one hand, desert dust can inhibit the NPF process by 1) reducing UV radiation and thus reducing photochemical processes and by 2) increasing the scavenging rates of precursor vapours by desert dust particles and thereby supressing or reducing the formation and growth rates during NPF events. By contrast, desert dust can also enhance NPF process by enhancing the formation of OH and other radicals by the presence of catalyst components in the desert dust particles, favouring oxidation reactions and therefore promoting the occurrence of NPF events during dusty conditions. In this sense, laboratory studies showed that the presence of $TiO_2$ and $Fe_2O_3$ (which are common components of desert dust) under UV light could promote the occurrence of NPF (Dupart et al., 2012). These components, acting as catalysts, are not consumed in the photo-catalytic reaction and can accelerate atmospheric photochemistry repeatedly. In fact, recent laboratory study (Zhang et al., 2023) revealed that $TiO_2$ contributes significantly to the formation of gaseous $H_2SO_4$ by increasing the GR and $J$ by up to a factor of 2 and 3, respectively, in the presence of $TiO_2$. Thus, in the following, the possible impact of desert dust on the characteristics and strength of NPF event occurred in the different studied sites was analysed.

Figure 4a shows the growth rates in the 10-25 nm size range at the different analysed stations for all the class I NPF events during the whole measurement period at each site and also for non-dusty, dusty and intense dusty days. The results for the whole measurement periods show that the GRs were in the same range of values at the 5 different studied sites, with mean





values of 5.3 nm h$^{-1}$ at IZO; 6.9 nm h$^{-1}$ at SNS; 6.2 nm h$^{-1}$ at MSY; 6.6 nm h$^{-1}$ at AMM and 6.5 nm h$^{-1}$ at HAS. These values agree with the GRs observed in previous studies at each studied site (García et al., 2014; Casquero-Vera et al., 2020; Carnerero et al., 2019; Hussein et al., 2020b; Hakala et al., 2019). When comparing the GRs obtained during dusty and non-dusty conditions, clear site-to-site differences can be observed. In this sense, the GR showed larger median values during dusty conditions than during non-dusty conditions at SNS (6.8 and 5.6 nm h$^{-1}$ for dusty and non-dusty conditions, respectively), MSY (10.1 and 4.8 nm h$^{-1}$) and HAS (6.2 and 5.2 nm h$^{-1}$). Slightly lower GR median values for dusty than for non-dusty conditions were observed at IZO (3.8 and 5.5 nm h$^{-1}$) and AMM (6.1 and 6.8 nm h$^{-1}$). In this sense, the previous analysis of the CS showed that the CS values during NPF events were similar or slightly larger for dusty days in comparison with non-dusty days (Fig. 3). However, this analysis of CS was done for the whole dataset, whereas GR was only retrieved for NPF class I events. Thus, in order to identify the possible role of CS on the growth rates, the CS has been retrieved for the corresponding NPF class I events during non-dusty, dusty and intense dusty conditions (Fig. 4b).

By looking at the relationship between GR and CS during class I events, the results show that the GR does not present a clear dependence on the CS at the studied sites during dusty and non-dusty conditions (Fig. 4 and S3). Nevertheless, the analysis of the corresponding CS during class I events shows that the CS values during dusty conditions were larger than those observed during non-dusty conditions at the IZO, MSY and HAS sites, and almost similar in both situations at the SNS and AMM sites (Fig. 4b). In this sense, at IZO, lower GRs were observed during dusty conditions in comparison with non-dusty conditions, a fact that could be explained by the increase of the CS during dusty conditions (Fig. 4) which can increase the removal of precursor vapours and therefore reducing the growth of newly formed particles. Lower GRs were also observed during dusty than during non-dusty conditions at AMM, but in this case there was a minimal difference (no statistically significant as confirmed by Mann-Whitney test at 0.01 significance level) in the value of CS between the dusty and non-dusty conditions. By contrast, larger GRs were observed at SNS, MSY and HAS during dusty conditions than during non-dusty conditions, but in these cases the values of CS were also larger during dusty conditions. Thus, the increase of the GR and CS during dusty conditions at these sites suggests that there was an increase of precursor vapours during dusty conditions or an increase of oxidation processes due to the presence of desert dust that could enhance the formation of condensing vapours that further promote the growth of particles. Furthermore, when considering dusty and intense dusty conditions, there were no statistically significant (confirmed by Mann-Whitney test at 0.01 significance level) differences in the values of GR at MSY, AMM and HAS, suggesting that even during the strongest dust episodes (with a significant increase of CS; Fig. 4b) NPF events occur and the growth rates do not differ from dusty conditions at these sites. However, the results show a reduction (statistically significant as confirmed by Mann-Whitney test at 0.01 significance level) in the GR values during intense dusty conditions at IZO and SNS, suggesting that despite NPF events can occurs under these intense episodes, the growth of new particles could be limited by the increase of CS due to the increase of precursor gases scavenging.





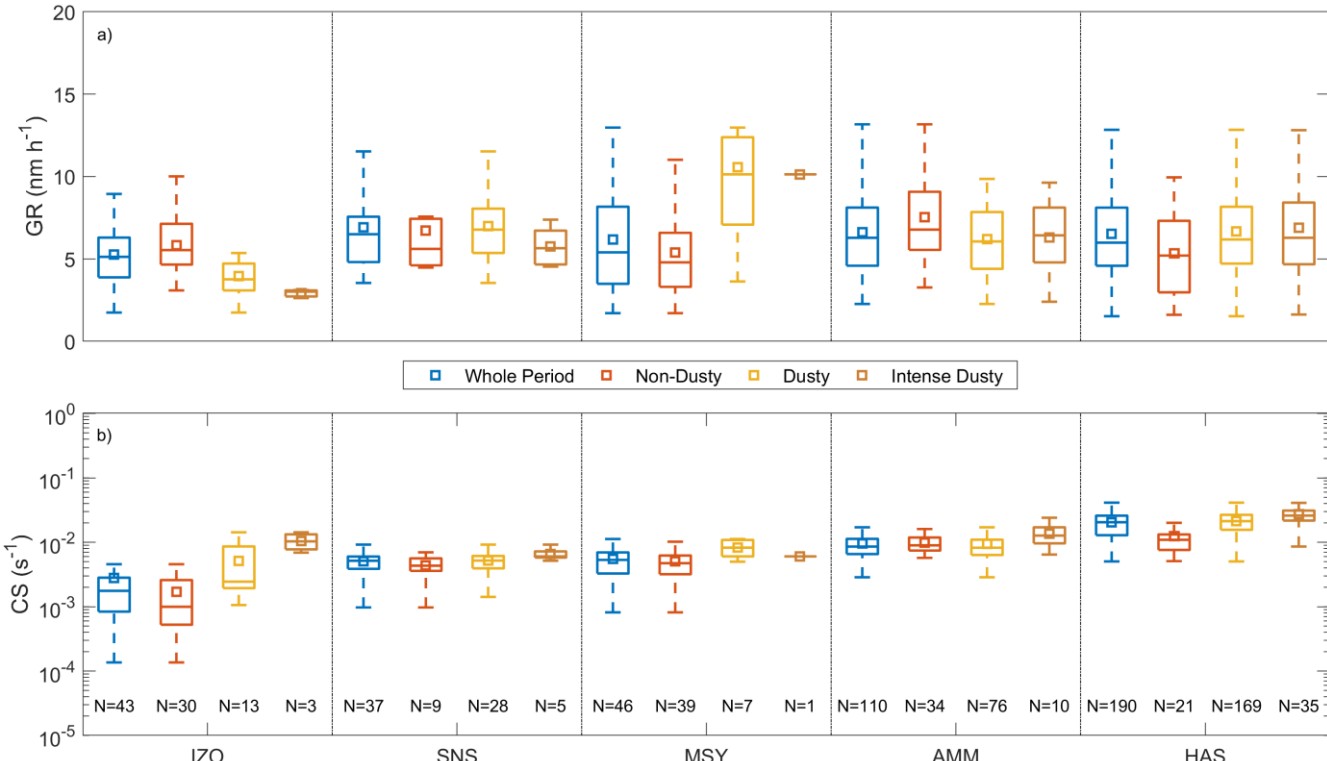

**Figure 4. Box and whisker plot of GR (10-25 nm) (a) and CS (b) recorded at each measurement station during all NPF class I events and for non-dusty, dusty and intense dusty conditions days. The line represents the median of the data, the square represents the mean of the data and the lower and upper edges of the box represent the 25th and 75th percentiles of the data, respectively. The whiskers go from the end of the interquartile range to the furthest observation within 1.5× interquartile range. N represents the total number of days for each category.**

Figure 5 shows the formation rates ($J$) for the overall class I events and also for dusty and non-dusty days. The results show that the lowest values of $J$ were observed for the high-altitude IZO site (0.38 cm$^{-3}$ s$^{-1}$), followed by the MSY remote (1.1 cm$^{-3}$ s$^{-1}$) and SNS high-altitude (1.3 cm$^{-3}$ s$^{-1}$) sites. By contrast, the highest values of $J$ were observed at AMM (1.8 cm$^{-3}$ s$^{-1}$) and HAS (3.8 cm$^{-3}$ s$^{-1}$), where also high values of CS were observed (Fig. 4b). $J$ shows a similar behaviour as CS from site to site, showing an increase from western to eastern sites. However, when comparing dusty and non-dusty conditions, the particle formation rates were almost similar at IZO (0.35 and 0.39 cm$^{-3}$ s$^{-1}$, for dusty and non-dusty conditions, respectively), SNS (1.2 and 1.3 cm$^{-3}$ s$^{-1}$) and AMM (1.7 and 1.8 cm$^{-3}$ s$^{-1}$). Larger particle formation rates were observed during dusty conditions at MSY and HAS sites (1.4 and 4.3 cm$^{-3}$ s$^{-1}$, respectively) compared with non-dust days (1.0 and 3.3 cm$^{-3}$ s$^{-1}$, respectively).

Thus, these results contrast with the results observed for the growth rates. For example, at AMM and IZO the observed GRs were lower during dusty days than in non-dusty cases, however, the formation rate at these sites shows similar $J$ values during dusty and non-dusty days. By contrast, at HAS and MSY the observed GRs are larger during dusty days and the same result is observed for the formation rate. These differences on the results for formation and growth rates point to the decoupling of





the mechanisms leading to the initial particle formation and the subsequent growth of the particles, and differences on these mechanisms during dusty and non-dusty days. Overall, the differences observed from site to site could be related to the differences on the origin and concentrations of precursor vapours, atmospheric conditions and chemical reactions during dusty conditions. However, the limited instrumentation available at the studied sites does not allow further investigation of the factors

involved in the increase/decrease of the GR and $J$ during dusty and non-dusty conditions and the link with the condensation sink. In fact, in presence of enough precursor gases, an increase of the growth and formation rate would be expected due to higher oxidation rates in the presence of catalyst components present in desert dust particles (Zhang et al., 2023).

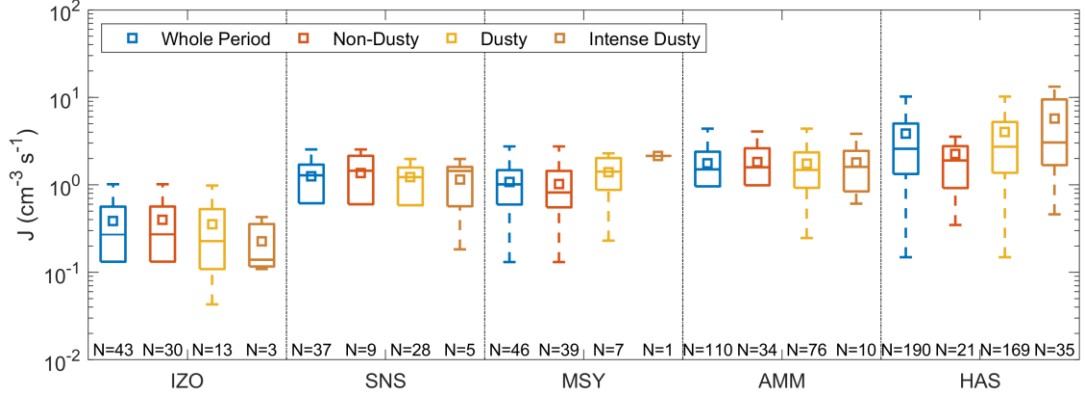

**Figure 5. Box and whisker plot of $J$ (10 nm) recorded at each measurement stations during all NPF events and for dusty and non-**
**dusty conditions days. The line represents the median of the data, the square represents the mean of the data and the lower and upper edges of the box represent the 25th and 75th percentiles of the data, respectively. The whiskers go from the end of the interquartile range to the furthest observation within 1.5× interquartile range. N represents the total number of days for each category.**

**3.5 Impact of desert dust outbreaks on particle concentration of CCN-relevant sizes**

Many modelling studies have suggested that NPF may impact on the abundance of the global CCN (e.g., Spracklen et al., 2006; Merikanto et al., 2009; Yu and Luo, 2009; Makkonen et al., 2012; Gordon et al., 2017), however the magnitude of the contribution of NPF to CCN still differs substantially between different models and for different environments. Additionally, desert dust particles are efficient CCN as well (e.g., Levin et al., 2005). Due to their large size, dust particles can also act as giant CCN (Feingold et al., 1999; Levin and Cotton, 2009) but according to Lee et al. (2009), CCN concentration decreases in

dusty regions up to 10–20% because dust competes for condensable $H_2SO_4$, reducing the condensational growth of ultrafine mode particles to CCN sizes. However, the results of this study reveal that NPF events can occur even during the strongest desert dust outbreaks, and that the growth and formation rates of new particles during dusty conditions can even increase at some studied sites. This could have a large impact on global CCN populations due to the large spatial coverage of dust. Since the impact of desert dust on NPF and, therefore, on CCN concentrations is still poorly understood, in this section we present

an analysis of the influence of desert dust outbreaks on the particle concentration of CCN relevant sizes.



In absence of direct CCN measurements, the CCN concentration can be estimated from continuous monitoring of the particle number size distribution. In this work, a simpler approach introduced by Lihavainen et al. (2003) was used. This indirect method has already been used in several studies (Asmi et al., 2011; Kerminen et al., 2012; Laakso et al., 2013; Laaksonen et al., 2005; Rose et al., 2017), and it is based on the assumption that all particles larger than a given activation diameter are

potential CCN. We considered 50 nm as the critical activation diameter (Asmi et al., 2011; Rose et al., 2017; Schmale et al., 2017) and, therefore, all particles larger than 50 nm were considered as potential CCN. This critical activation diameter is consistent with previous studies based on direct CCN measurements, which indicate that the smallest particles involved in the formation of real atmospheric cloud droplets are usually in the range of 50–150 nm, depending on supersaturation, SS (e.g., Kerminen et al., 2018). In fact, Rejano et al. (2023) observed critical diameters of 50-60 nm at SS=0.75% for different aerosol

sources and, therefore, $N_{50}$ described below was considered a good CCN proxy at high SS values.

In order to estimate CCN concentrations at the different studied sites, median particle number concentrations of particles larger than 50 nm in diameter were retrieved for each day at each measurement site in the 15:00-18:00 h time window ($N_{50}$ from now on; average time window at which particle growth from NPF events taper off; Fig. S4). The overall results show that during the study period, the values of $N_{50}$ range over an order of magnitude, with low median $N_{50}$ values of 238, 1358 and 2225 $cm^{-3}$ observed

at western sites at IZO, SNS and MSY, respectively. Higher median $N_{50}$ values of 4532 $cm^{-3}$ was observed at HAS remote arid site and due to the urban characteristic of the AMM site and the larger influence of anthropogenic emissions, the largest median concentration was observed at AMM with a median value of 5417 $cm^{-3}$. These results show again significant differences between sites in $N_{50}$ as those observed in Sect. 3.3 for the CS.

When comparing the observed $N_{50}$ between dusty and non-dusty conditions, a reduction of 28% from the non-dusty to dusty

conditions was observed at the SNS site (Fig. S5). The relatively lower NPF occurrence during the dusty conditions (66%) in comparison to that observed in non-dusty days (85%) at this site (Fig. 2 and 7), and/or the increased scavenging of fine particles and precursor gases during dusty conditions, may explain the reduction of $N_{50}$ on dusty days at the SNS site. No statistically significant differences (Mann-Whitney test at 0.01 significance level) in $N_{50}$ were observed between the dusty and non-dusty conditions in the other studied sites. At these sites, the frequency of NPF events occurring on dusty conditions is similar or

even slightly higher than the observed during non-dusty conditions (Fig. 2), which may explain the small differences in $N_{50}$ during dusty and non-dusty conditions at these sites. In the following, the role played by NPF during desert dust periods on CCN will be investigated.

NPF events are one of the main processes producing aerosol particles in the size ranges at which particles could act as CCN (e.g., Kerminen et al., 2012). However, NPF events only contribute to CCN budget at an specific site when NPF events present

a subsequent growth lasting for hours and reaching climate relevant sizes for CCN formation, it means during class I events (e.g., Rose et al., 2017; Rejano et al., 2021). Therefore, although NPF events occur frequently at the studied sites, this does not necessarily imply the growth of newly formed particles into CCN-relevant sizes at the sites. Figure 6 shows the percentage of




NPF class I events that occurred during the whole measurement period and for non-dusty, dusty and intense dusty conditions. As can be seen, 39% and 24% of the total number of NPF events observed during the whole analysed period in IZO and MSY, respectively, were of class I (Fig. 6). The maximum sizes that particles reached during the class I NPF events at IZO site were below 30 nm (Fig. S4), and thus we expect that the impact of NPF events on the CCN budget at this site will be very limited.

In contrast, the particles formed during the class I NPF events at MSY site reached sizes larger than 50 nm (Fig. S4), and thus these events can impact CNN budget over this site. On the other hand, 70%, 100% and 97% of the total number of NPF events observed during the whole analysed period in SNS, AMM and HAS, respectively, were of class I (Fig. 6). Considering the high percentage of NPF class I events occurring at SNS, AMM and HAS, and the larger sizes that the particles reached during these NPF events (> 60 nm; Fig. S4), a significant impact of NPF events on CCN budget at these sites is expected.

Similar to our previous results, the differences in the percentage of class I NPF events between the dusty and non-dusty conditions depends on site. In this sense, the percentages of class I events observed during the non-dusty conditions at SNS and MSY sites were 82% and 26%, respectively, and slightly decreased to 67% and 15%, respectively, during the dusty conditions (Fig. 6). By contrast, the percentages of class I NPF events observed at IZO and HAS during the dusty conditions (42% and 97%, respectively) were slightly larger than those registered during the non-dusty conditions (38% and 95%, respectively). However, differences

were larger when considering the intense dusty conditions, showing a reduction of the class I NPF events percentage at all the sites, except at IZO and AMM. In this sense, it is worth to mention that at IZO the increase of the event frequency was due to the low number of NPF events during intense dusty conditions, with just 4 NPF events during those conditions. Thus, despite there was a reduction on the event frequency, the similar maximum size that particles reached during NPF events during dusty and non-dusty conditions (Fig. S4) suggest that the NPF events can also significantly contribute to the CCN budget during dusty conditions.

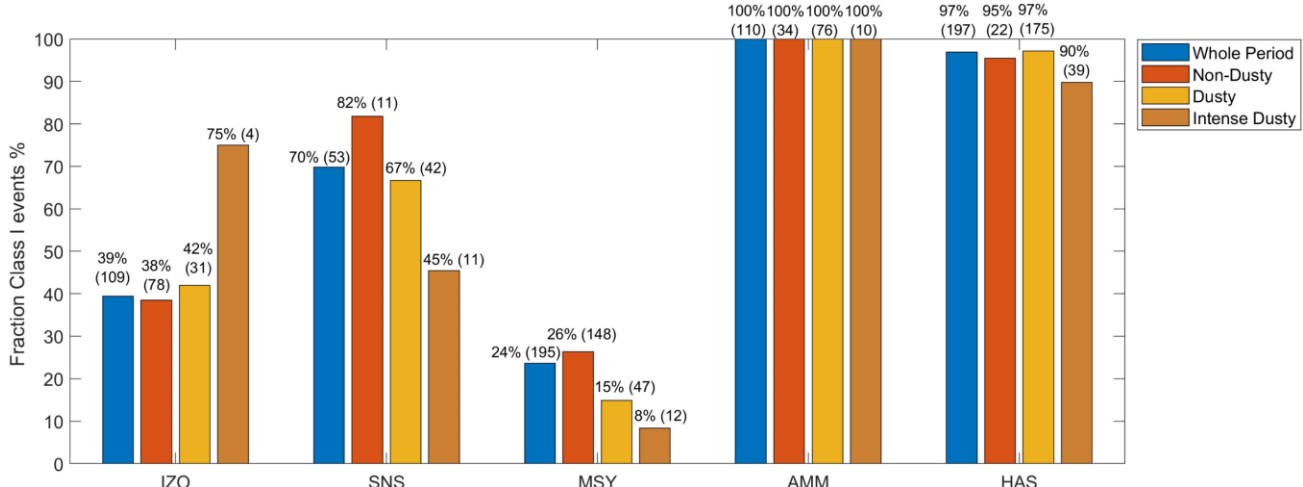

**Figure 6. Percentage of class I NPF events for the whole period and non-dusty, dusty and intense dusty conditions at the 5 studied sites. In parenthesis are the total number NPF events for each category.**





Figure 7 shows $N_{50}$ during all NPF classes (E), class I NPF events and non-NPF events (NE) for dusty and non-dusty conditions. When comparing NE days during dusty and non-dusty conditions, the results show that $N_{50}$ values were statistically significantly (Mann-Whitney test at 0.01 significance level) lower during dusty conditions at all the sites, except at IZO. This reduction in the potential CCN budget during NPF events under dusty conditions suggests an increase of the coagulation rate during dusty conditions. This result partly agrees with the modelling results presented by Karydis et al. (2017), who suggested that desert dust enhances the CCN budget over cleaner areas, whereas, in polluted regions, desert dust can deplete the CCN budget.

When comparing E and NE days during dusty and non-dusty conditions, $N_{50}$ showed larger values during the E days than during the NE days for both dusty and non-dusty conditions, except at AMM (Fig. 7), which suggests that NPF events contribute significantly to the CCN budget during both dusty and non-dusty conditions. Furthermore, larger differences in $N_{50}$ between the E and NE days were observed during the dusty conditions than during the non-dusty conditions at SNS, HAS and MSY, suggesting that NPF during desert dust episodes has larger impacts on CCN at these sites. Assuming that NE and E days have the same aerosol background conditions, this result suggests that the contribution of NPF events to CCN budget is larger during dusty conditions than during non-dusty conditions. In addition, the $N_{50}$ differences between the dusty and non-dusty conditions were even larger when we only consider class I NPF events (Fig. 7). During both dusty and non-dusty conditions, $N_{50}$ during the class I NPF events showed larger or similar concentrations than during the E and NE days, suggesting that, as expected, class I NPF events contributes significantly to the CCN budget. In this sense, during the dusty conditions, $N_{50}$ showed an increase from 274 (for NE days) to 356 cm$^{-3}$ (for class I NPF days) at IZO, from 662 to 1516 cm$^{-3}$ at SNS, from 1156 to 2738 cm$^{-3}$ at MSY and from 3342 to 4989 cm$^{-3}$ at HAS. By contrast, AMM was the only site where the class I NPF event days presented lower $N_{50}$ concentrations than on NE days, showing a reduction from 5706 cm$^{-3}$ during NE days to 4847 cm$^{-3}$ during class I NPF event days. Thus, these results suggest that, during dusty conditions, the potential CCN budget is at least 1.5 times larger (except at AMM) during the class I NPF events than on NE days. This means that we could be underestimating the CCN budget by 50% if we do not consider the occurrence of NPF events during mineral dust outbreaks.

Estimating the contribution of NPF events to CCN budgets is challenging. In the literature, several approaches have been described to estimate the contribution of NPF to CCN, however no clear agreement between the different methodologies exist yet (e.g. Dameto de España et al., 2017; Rose et al., 2017; Kalkavouras et al., 2019). The main difficulty of these approaches arises from the fact that different aerosol sources other than NPF might be contributing to the CCN population simultaneously. In addition, stationary observations limit the observation of NPF events to a certain time frame and, therefore, particles growth towards larger sizes cannot be neglected, contributing to CCN budget far from the measurement site. Moreover, during the particle growth to CCN sizes, aerosol properties may change because of changes in the meteorological conditions (change of air mass or mixing layer height) or because of influences by other aerosol sources. Thus, despite in this work we observe an increase on the potential CCN population at the studied sites during NPF event days, further efforts are needed to estimate the contribution of NPF events to the global CCN population during dusty and non-dusty conditions.



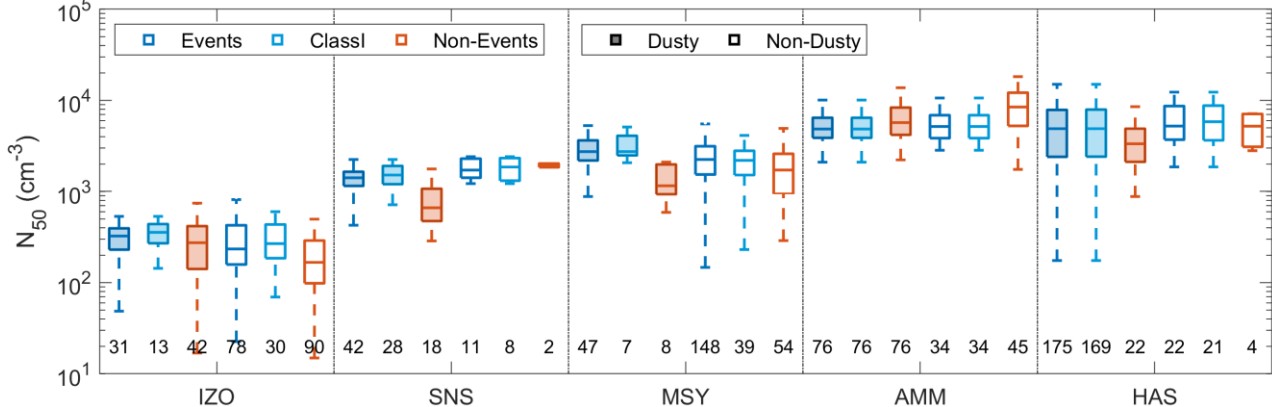

**Figure 7. Box and whisker plot of $N_{50}$ recorded at each measurement site during NPF events (blue), class I events (light blue) and non-events (red) and during dusty (shaded boxes) and non-dusty (non-shaded boxes) condition days. The line represents the median of the data and the lower and upper edges of the box represent the 25th and 75th percentiles of the data, respectively. The whiskers go from the end of the interquartile range to the furthest observation within 1.5× interquartile range. Numbers on bottom represent the total number of days for each category.**

While we provide an estimation of the impact that desert dust outbreaks have on number concentrations of particles larger than 50 nm, several questions remain unexplained, such as how desert dust influences cloud formation. In this sense, desert dust has been thought to uptake condensable material (such as sulphuric acid or organics) onto the surface of desert particles, depleting the reservoir of material required to create other CCN. However, we found that NPF events occur during dusty conditions, implying that NPF events contribute to CCN budgets at the studied sites even during dusty conditions. Thus, in addition to the possible particle coating by soluble material which could increase CCN activity of desert dust particles, the formation and growth of new particles is an additional source to be taken into account. However, all this would depend on the origin of desert dust and precursor vapours. Thus, to improve our understanding in the effect of desert dust outbreaks on NPF and CCN budget, further investigation accompanied by multiplatform measurement campaigns with state-of-the-art gaseous and particulate physical and chemical properties measurements is still needed.

## 4 Summary and conclusions

We investigated the occurrence and characteristics of new particle formation (NPF) events under the influence of desert dust outbreaks in 5 different dust-influenced areas. Unexpectedly, our results show that the occurrence of NPF events is highly frequent during desert dust outbreaks, showing that NPF event frequencies during dusty conditions are similar to those observed during non-dusty conditions. Furthermore, our results show that NPF events also occur during intense desert dust outbreaks at all the studied sites, even at remote sites, pointing out that NPF events occurrence is not only limited to highly polluted dust plumes, occurring in remote sites during intense mineral dust outbreaks. However, during desert dust outbreaks, and especially during intense desert dust episodes, there is a reduction in the frequency of class I NPF events at some of the studied sites.

Usually, the condensation sink (CS) is calculated by only considering the number size distribution of fine mode particles. However, our results show that in dust-influenced areas, the value of CS can be considerably underestimated if coarse mode particles are not taken into account, with up to 17% underestimation in the studied sites. In fact, the condensation sink associated with coarse particles ($CS_C$) may represent up to 60% of the total CS during dusty conditions in clean environments,

and that the $CS_C$ obtained during dusty conditions is 1.5 to 20 times larger than the observed during non-dusty conditions at the studied sites. However, despite the large impact of desert dust on total CS, our results show a dependence from site to site on the effect of desert dust outbreaks on the suppression/promotion of NPF events.

We did not find a clear pattern of the effect of desert dust outbreaks on the strength of NPF events (quantified by the particle growth and formation rates). In this sense, looking at the relationship between the growth rate (GR) and CS during class I NPF

events, the results show that the GR does not present a clear dependence on CS at the studied sites during dusty and non-dusty conditions, and that the effect of CS and desert dust on the GR vary from site to site. These differences could be related to 1) an increase/reduction of precursor vapours with the increased/reduced transport of pre-existing particles and/or 2) the chemical composition of pre-existing particles and precursor vapours, that can produce differences in the scavenging efficiency of CS and its inhibiting effect on NPF events.

In addition, there are different effects of desert dust outbreaks on the particle growth and formation rates at each site, suggesting different formation and growth mechanisms during dusty and non-dusty days. However, the differences observed from site to site could be also related to the differences on precursor vapours origin and concentrations, atmospheric conditions and chemical reactions during dusty conditions. However, the limited instrumentation available at the studied sites does not allow further investigation of the factors involved in the increase/decrease of the GR and $J$ during dusty and non-dusty conditions

and the link with the condensation sink.

Finally, despite desert dust has been thought to uptake condensable material onto the surface of desert particles (depleting the reservoir of material required to create other CCN), we found that NPF events occur during dusty conditions, implying that NPF events can contribute to CCN budget even during dusty conditions. In this sense, we found that when class I NPF events occur during dusty conditions, newly formed particles reach similar maximum sizes than during non-dusty conditions. However, the

effect of desert dust on NPF and CCN budget would depend on the origin of desert dust and the origin and concentration of precursor vapours. Thus, due to the foreseeable increase of the frequency, duration, and intensity of desert dust events due to climate change, it is imperative to improve our understanding of the effect of desert dust outbreaks on NPF and CCN budget with further investigation accompanied by multiplatform measurement campaigns and chamber experiments with state-of-the-art gaseous and particulate physical and chemical properties measurements is still needed.



**Data availability**

The data used in the manuscript is available from the first author at casquero@ugr.es or juan.casquero@helsinki.fi.

**Author contributions**

JACV performed the data harmonization, treatment and formal analysis and wrote the manuscript. JACV, DPR and HL carried
out the conceptualisation and investigation and, together with GT, they carried out a thorough proofreading of the manuscript
before obtaining the final version of the manuscript. FR, AC, GT, LD, SH, TH, PP, KL, TP, FJO and LA assisted in the
conceptualisation. SR, JACV, NP, TH, SH and AH provided the datasets for the different measurement sites. All authors
contributed to the discussion of the results and provided comments on the paper.

**Competing interests**

At least one of the (co-)authors is a member of the editorial board of Atmospheric Chemistry and Physics.

**Financial support**

Juan Andrés Casquero-Vera is funded by FJC2021-047873-I funded by MCIN/AEI/10.13039/501100011033 and
NextGenerationEU/PRTR, by Spanish Ministry of Universities and the European Union – NextGenerationEU and by the
Academy of Finland through ACCC Flagship (Atmosphere and Climate Competence Center, project no. 337549). This
research has been partially supported by the Spanish Ministry of Science and Innovation (grant nos. FJC2021-047873-I,
PID2020-12001-5RB-I00, RED2022-134824-E and PID2019-108990RB-I00) and by the European Union's Horizon 2020
research and innovation program through projects ACTRIS IMP (grant no. 871115) and ATMO ACCESS (grant no.
101008004) and via Horizon Europe through Non-CO2 Forcers and their Climate, Weather, Air Quality and Health Impacts,
FOCI (project no. 101056783). This research was also partially supported by Plan Propio of University of Granada through
EMERALD project (PPJIA2022-15) and the programs Singular Laboratory AGORA (LS2022-1) and Scientific Units of
Excellence Program (grant no. UCE-PP2017-02) programs, by the Junta de Andalucía Excellence projects ADAPNE (P20-
00136) and AEROPRE (P-18-RT-3820) and the Consejo Superior de Investigaciones Científicas (CSIC) under the Project
202030E261. Measurements at Hada Al Sham were funded by the Deanship of Scientific Research (DSR, grant no I-122-430)
at King Abdulaziz University (KAU).



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
