# Peer review of "Impact of desert dust on new particle formation events and cloud condensation nuclei budget in dust-influenced areas"

_EGUsphere, 2023_

## Referee Comment (RC2)

REview : Impact of desert dust on new particle formation events and cloud condensation nuclei budget in dust-influenced areas.

This paper is reporting NPF event observed at 5 different sites over the Mediterranean coast. The authors sorted the NPF events observed during non-dusty, dusty and intense dusty days and then compare different parameter such as N50, Cs, GR, J. Nowadays, many previous papers have highlighted the fact that the preexisting particles, such as in polluted areas, are not necessarily blocking the NPF occurrence. In this manuscript, the authors claim that dust events are associated with larger number of pre-existing particles that doesn't prevent NPF events. According to me, this manuscript needs major revision before publication.

General comments :
- The authors' choice of colors does not facilitate the reader's understanding because the differences are very subtle. Moreover, nowadays there are some tools to help the authors to choose the color scheme specifically for color blinded readers.

- All sites cited in this paper are extremely different. Some are within the free troposphere most of the days and some and in rural or urban environment. Therefore, the processes to form new particles are extremely different. NPF are highly influenced by the environmental conditions such as RH, radiation, temperature, cloud fraction etc… It is not stated here what are the difference from site to site and from event to event for each particular site. Do you know what are the main environmental factor influencing the NPF over all the sites independently ? Maybe it needs to be constrain for further comparison. Choosing only the events occurring at RH lower than 60%, or at high radiation (larger than a threshold), to avoid any influence of the environmental conditions. Moreover, I believe that most of the NPF event are strongly influenced by the dynamic of the atmosphere (BL height). Within the FT, the BLH is dragging enough vapors to condense while in an urban site that would disperse a bit the preexisting particles. I think this is important to discuss that in the manuscript. For exemple, How often are IZO and SNS within the BL ? Are all the events occurring within the BL or just when the BLH is reaching the station ?

- I have issued with the methodology used here. First, the authors stated that the Dust event are characterized using simulation results and confirmed with OPS/APS data from each station (P7 L20). I think it would be valuable to show the $N_{200}$ or the PM10/PM1 during non-dusty, dusty and intense dusty days. Indeed, the dust SD could be different from event to event but I guess most of the dust diameter would be, as you suggested, larger than 200nm. Does the Dust event remains the whole day for all cases ?
Moreover, as it is not clearly stated, I believe that the authors selected all the values observed during event and non-event days as well as during non-dusty, dusty and intense dusty days and compare it as is. However, all those NPF events can have a strong impact on CS, $N_{50}$ or $N_{200}$. If you want to use the CS as a limiting parameter for NPF occurrence, could you please compare the CS before the NPF event start? It's usually called the CS2 (2 hours prior to the event) and it would be much more interesting than using tall the daily recorded values CS. Indeed, the SD clearly show that the particles are not growing to the same size. Therefore, some newly formed particles will have a clear effect on the CS (diameter reaching 50 or 60 nm) while the newly formed particles that remained below

30nm won't have the same effect. That would lead you to a biased interpretation. Same advices for $N_{50}$ : You are trying to better understand the impact of those NPF events on the $N_{50}$. Then to do so I would compare $N_{50}$ before the event starts and after the event starts. Indeed, the $N_{50}$ seems to be similar during NPF Event/NonEVENT/Dusty/nonDusty. As presented right now it's hard to understand if the N50 differences are due to the NPF events or the dust events.

- As it is not stated clearly, I believe that the CS was calculated assuming that the condensable vapors have molecular properties similar to sulfuric acid and therefore this parameter is not "only" depending on the aerosol size distribution as stated by the authors.

Comments :

P3 L5 : "only the particle number size distribution is considered." Again It's not entirely true…

P8 L 19 "the highest frequency was observed at HAS." Are you not discussing SNS results in purpose ?

P9 L11_15 : I found it weird to read (50% of the paragraph) about seasonal variability within a paragraph where you state it has no seasonal variability…

P10 L 14/15 : Not well said : "The results agree with the periods "

P10 L21 : Why not showing the CSc and CSf ?

P11 L 5-6 : "Therefore, the occurrence of NPF events at these sites is probably not only limited to highly polluted dust plumes (as suggested by Nie et al., 2014), and they can even occur in remote sites during desert dust outbreaks." As stated just above this is true only if there is enough vapours to condense !

P11 L9-10 : "Thus, this result evidences that the intensity of desert dust can limit the occurrence of NPF events at IZO and AMM, which could be explained by a significant reduction or a limited amount of precursor gases to compete with the increase of available surface of pre-existing particles. " So your hypothesis is that there are no condensable vapors within the dusty air mass ? IZO is within the FT or the BLH at that time ? At AMM, urban site, a lot of condensable vapors are available and when the dusty air masses are coming that will increase the number of preexisting particles and would block the process? I found it difficult to state that in one sentence without proof. That would be interesting to focus on those days to better understand the role of the environmental parameters known to play a large role such as RH, radiation etc…

Figure 2 :Can you change the way you plotted it so it could help the reader? Please add some texture for event, Non event and undefined days ?
The colors for non Dusty/dusty and Intense dust are really close. Can you pick another color ?

P12 L1-3 : You have said that multiple times and this is I think not necessary..

P12 L25 and P12 L30 one more repetition. I believe that this is not 21 times more in IZO but just 2 times more ! Again here I would use the CS2 so you can actually use it as a parameter that could prevent or not the NPF event.

P12 L31 : You can't say that the CS is increasing just by the removal of fine particle through coagulation without proving it. Could you plot the CS as a function of the diameter at different moment of the day (before /during and after the NPF event). From here you could state this strong conclusion supported by results : "Thus, these results highlight the impact of desert dust outbreaks into the CS and the importance of considering coarse mode particles for adequate CS calculations at desert dust-influenced areas."

P13 : Here I think that all the explanations are dubuious since you used all CS values recorded during the day. I would strongly recommend to recalculate the CS 2 hours prior to the NPF events.

P14 L12-14 : This should probably go within the introduction such as the sentence from P14 L19-21. I would appreciate having more information about the TiO2 impact.

"In fact, recent laboratory study (Zhang et al., 2023) revealed that TiO2 contributes significantly to the formation of gaseous H2SO4 by increasing the GR and J by up to a20
factor of 2 and 3, respectively, in the presence of TiO2."
TiO2 is in the particular phase. How could it influence the production of H2SO4 ? You mean that there are more H2SO4 because the GR increase ? So there is a particular chemistry that would enhance the condensation rate of some vapors (which ones ??? ) and in the same time there will be more H2SO4 within the vapor phase ?

P15 – L13-15 : That was expected to have stronger values of CS during dusty days. Again, I would present here CS2 so you do not take into account the NPF event influence on the GR …

Figure 4 : At MSY you have only one event during intense dust days. However, from figure 2 you have at least 10 NPF events. Please help us to understand… From what I understood the whole period here corresponds to events during non dusty right ? It can't be the dusty and non dusty NPF events otherwise the number would be much larger ! Change the legend to state it clearly what it referes to … .
But then if this is events during non dusty days it cshould correspond to the numbers on Figure 3 and 7 which is not…

P18 L9. A SS of 0.75% is an extremely large value and correspond to very very fast-growing clouds. I'm not sure these SS are observed in reality. That means that you choose an activation diameter that is really low. So ou are probably overestimatin the number of CCN available with this threshold. It needs to be stated somewhere.

Section 3.5 again here I would plot the N50 before and after the event start to clearly separate the impact of the newly formed particles on the N50. So you could estimate the impact of NPF during dusty days in comparison to non-dusty days .

P19: Do you know why the newly formed particles do not grow to larger size ? It seems from Figure S4 that there is a threshold different from site to site for the diameter reached by the end of the NPF events.
Figure S4 is an average over dusty and non-dusty days right ? It seems that SNS show more large particles (around 100nm) during non dusty days. Why is that ?  I don't see  a clear difference for AMM. Can you comment ? IZO the SD does not show a banana shape. So I'm not sure it could be sorted as class I event. Can you comment on that ?

Figure 6 : I would not show this figure but I would add a table with those numbers. Maybe you can  insert it in Table 1.

P20 and Figure 7 : According to Figure7 there are no statistical differences between the Non dusty and dusty days for N50 at IZO, AMM and HAS. Indeed, the boxplots show similar 0.25 and 0.75 percentile behaviors for these 3 sites. So this is hard to draw strong conclusions on the effect of dusty events on the NPF efficiency to increase N50.

Again add texture instead of just having a light blue and blue colors...

The $N_{50,dusty}$ is sometimes lower than $N_{50,non-dusty}$ (SNS, AMM and HAS ). So first I'm not sure I'm able to understand that especially since you stated that you selected the dust evet by using the SSD of the coarse mode. Now, supposedly you have a dust event for the whole day (it's not stated what is the duration of these dust events). So the $N_{200}$ should be higher during dusty events (need to show that to draw the later conclusions).  How is the $N_{50-200, non-dusty}$ in comparison to $N_{50-200, dusty}$ ? Again as some newly formed particles grow larger than 50 nm I can't tell if this is due to the increase is solely coming from the NPF event during non-dusty days that could lead to larger particles due to more vapors available to grow. I strongly advice to compare the increase of N50 before in comparison to after the NPF events and find a way to normalized it according to the dust concentration so we can clearly understand the effect of NPF/Dust on the CCN concentration.

---

## Author Comment (AC1)

**Review of Casquero-Vera et al. "Impact of desert dust on new particle formation events and cloud condensation nuclei budget in dust-influenced areas" by Anonymous Referee #1**

**We thank the reviewer for his/her valuable comments and suggestions that helped us to improve the quality of the manuscript. Our responses to the reviewer's comments are detailed below. Our answers to reviewer are shown in bold and the changes inserted in the manuscript are noted here in italic and between quotation marks. The changes in the new version of the manuscript are noted in red.**

**Response to referee #1:**

The paper presents a multi-site analysis of the occurrence of NPF events under the presence of desert dust particles in dust-influenced areas. In the paper, authors characterize NPF events at 5 different locations highly influenced by desert dust outbreaks under dusty and non-dusty conditions by using continuous measurements of aerosol size distribution in both fine and coarse size fractions. In this study, results show that the occurrence of NPF events is highly frequent during desert dust outbreaks, showing that NPF event frequencies during dusty conditions are similar to those observed during non-dusty conditions. Furthermore, results showed that NPF events also occur during intense desert dust outbreaks at all the studied 5 sites, even at remote sites where the amount of precursor vapours is expected to be low. Furthermore, authors also found that the contribution of NPF events to cloud condensation nuclei (CCN) budget is larger during dusty conditions than during non-dusty conditions. This study is imperative to improve our understanding on the effect of desert dust outbreaks on NPF and CCN budget for better climate change prediction.

Overall, the study describes the background and introduction, and methods in a comprehensive way. Therefore, I would encourage the authors to submit a revised manuscript by addressing my specific comments below:

**A point by point response is included below.**

1) Please explain Figures 5 and 7 in detail, more explicitly, So far I don't understand the statement in the current manuscript "....Thus, in addition to the possible particle coating by soluble material which could increase CCN activity of desert dust particles, the formation and growth of new particles is an additional source to be taken into account. However, all this would depend on the origin of desert dust and precursor vapors...".- please be clear about what consequence is the authors referring to.

**It is not clear to the authors what the reviewer refers to, but we have revised the section related with Figure 7 and the following paragraph has been modified as follows (P21 L20-P22 L6):**

*"While this study reveal a significant impact of class I NPF events on CCN budget during desert dust outbreaks at the studied sites, several questions remain unexplained, such as 1) how desert dust influences cloud formation?, and 2) what is the separate*

5 *contribution of desert dust and NPF to the CCN budget during NPF occurring during desert dust periods?. In this sense, desert dust has been thought to uptake condensable material (such as sulphuric acid or organics) onto the surface of desert particles, depleting the reservoir of material required to create other CCN. However, we found that NPF events occur during dusty conditions, implying that NPF events contribute to CCN budgets at the studied sites even during dusty conditions. Thus, two factors could increase the CCN budget during desert dust events: 1) the possible particle coating by soluble material*

10 *which could increase CCN activity of desert dust particles and 2) as shown here, the formation and growth of new particles to CCN sizes which is an additional source to be considered. However, all this would depend on the origin of desert dust and precursor vapours. Thus, to improve our understanding in the effect of desert dust outbreaks on NPF and CCN budget, further investigation accompanied by multiplatform measurement campaigns with state-of-the-art gaseous and particulate physical and chemical properties measurements is still needed."*

15

2) In Page 6 lines 3-4: "The classification of NPF event days was done by visual inspection of the daily particle number size distribution data according to the guidelines presented by Dal Maso et al. (2005). "- what are the changes which relate to this study?

**The criteria used here is the same than the one used by Dal Maso et al. (2005) for the classification on event, non-event,**
20 **undefined and bad-data days. However, Dal Maso et al. (2005) classified events on class I and II events and further divided class I events into sub-classes Ia and Ib. Dal Maso et al. (2005) refers to class Ia as those events with clear with strong particle formation events and little or no pre-existing particles, making them suitable for modelling case studies and class Ib contains the rest of class I events. Our classification uses the same criteria than Dal Maso et al. (2005) but including class Ib events into the class II category (events when the NPF growth rate retrieval is not possible) and the**
25 **events called by Dal Maso et al. (2005) as Ia (when the NPF growth rate retrieval is possible), are simply label in this study as class I events.**

**In order to clarify this statement we have modified the paragraph as follows:**

*"The classification of NPF event days was done by visual inspection of the daily particle number size distribution data according to the guidelines presented by Dal Maso et al. (2005). According to this classification criteria, days are classified*
30 *into four groups: NPF event (E), non-event (NE), undefined (UN) and bad-data days (BD). (1) "E" days are days during which sub-25 nm particle formation and their consequent growth are observed, (2) "NE" days are days on which neither new*

*growing modes nor production of sub-25 nm particles are observed, (3) "UN" days are the days which do not fit either of the previous classes, and (4) "BD" days are the days during which data are not valid or inexistent. In addition, event days are separated into two different groups: class I and II events. Class I contains days with very clear and strong new particle formation in which it was possible to retrieve the formation and growth rate and class II includes the rest of NPF events."*

3) In Page 12 lines 2-4: This complicated sentence is too long to understand.

**This sentence has been removed in this version of the manuscript because it is information that has already been mentioned before (e.g., P2 L26-27).**

4) What is the unit of the condensation sink (CS)?

**The unit of the condensation sink (CS, CSc and CSf) is "$s^{-1}$" and first time we present the unit is P12 L21 and figure 3.**

5) In introduction section, please be clear about the specific current situation and problems.

**We have modified the introduction including the lack of knowledge on the effect of the CS (P3 L4-8) and also a discussion about why desert dust acting as CS is of special interest and the role that some desert dust chemical compounds could have on the process of NPF events (P3 L17-29):**

**P3 L4-8:** *"However, traditionally, the effects of the morphology, physical state and chemical composition of the pre-existing particles are ignored in the calculation of CS. Overall, according to theoretical calculations, CS still too high for NPF events to occur (Du et al., 2022) and, therefore, further investigations on the effect of different vapours and pre-existing particle characteristics on the occurrence of NPF events are still needed (Tuovinen et al., 2020; 2021)."*

**P3 L17-29:** *"The climatic effects of desert dust and atmospheric NPF have been thought to be disconnected from each other, however, high dust loadings can affect NPF in opposing ways. Desert dust can inhibit the NPF process 1) by reducing UV radiation and thus reducing photochemical processes and 2) by increasing the scavenging rates of precursor vapours by desert dust particles and thereby supressing or reducing the new particle formation and growth rates during NPF events (de Reus et al., 2000; Ndour et al., 2009). However, desert dust can also enhance the NPF process by enhancing the formation of OH and other radicals by the presence of catalyst components in the desert dust particles, favouring oxidation reactions and therefore promoting the occurrence of NPF events during dusty conditions. In fact, several authors revealed that $TiO_2$ and $Fe_2O$ (which are common components of desert dust) act as photocatalysts and under UV light could promote the heterogeneous oxidation of $SO_2$ and the subsequent formation of gaseous $H_2SO_4$, inducing NPF (e.g. Dupart et al., 2012 and references therein). These*

*components, acting as catalysts, are not consumed in the photocatalytic reaction and can accelerate atmospheric photochemistry repeatedly. Furthermore, more recent laboratory study showed that the presence of $TiO_2$ greatly promotes NPF and can enhance the particle formation and growth rates by a factor up to 3 and 2, respectively (Zhang et al., 2023). However, clear association between desert dust loadings and the occurrence or strength of NPF has not yet been established.*"

---

## Author Comment (AC2)

**Review of Casquero-Vera et al. "Impact of desert dust on new particle formation events and cloud condensation nuclei budget in dust-influenced areas" by Anonymous Referee #2**

5 **We would like to acknowledge the work done by the referee in the revision of our manuscript. Our responses to the reviewer's comments are detailed below. Our answers to reviewer are shown in bold and the changes inserted in the manuscript are noted here in italic and between quotation marks. The changes in the new version of the manuscript are noted in red. A point by point response is included below.**

Response to referee #2 general comments:

10 This paper is reporting NPF event observed at 5 different sites over the Mediterranean coast. The authors sorted the NPF events observed during non-dusty, dusty and intense dusty days and then compare different parameter such as N50, Cs, GR, J. Nowadays, many previous papers have highlighted the fact that the preexisting particles, such as in polluted areas, are not necessarily blocking the NPF occurrence. In this manuscript, the authors claim that dust events are associated with larger number of pre-existing particles that doesn't prevent NPF events. According to me, this manuscript needs major revision before publication.

15 **We agree that previous studies have shown that NPF events occurs in very polluted areas with high pre-existing anthropogenic particle concentrations, but also with high levels of precursor vapours. However, the results obtained in this study show that NPF events also occur during intense desert dust outbreaks even at remote sites (e.g. SNS and IZO) where the amount of precursor vapours is expected to be low. In any case, the current knowledge about the effect of CS on the process of NPF is still poorly understood and the observed CS is still too high for NPF occurrence according**
20 **to theoretical calculations (Du et al., 2022). Therefore, further investigations on the effect of different vapours and pre-existing particle chemical composition on CS and NPF events is still needed (Tuovinen et al., 2020; 2021).**

1) The authors' choice of colors does not facilitate the reader's understanding because the differences are very subtle. Moreover, nowadays there are some tools to help the authors to choose the color scheme specifically for color blinded readers.

25 **Following reviewers' suggestions, we have changed the colours using colour blind friendly palette and added texture to some of the figures.**

2) All sites cited in this paper are extremely different. Some are within the free troposphere most of the days and some and in rural or urban environment. Therefore, the processes to form new particles are extremely different. NPF are highly influenced

by the environmental conditions such as RH, radiation, temperature, cloud fraction etc… It is not stated here what are the difference from site to site and from event to event for each particular site. Do you know what are the main environmental factor influencing the NPF over all the sites independently ? Maybe it needs to be constrain for further comparison. Choosing only the events occurring at RH lower than 60%, or at high radiation (larger than a threshold), to avoid any influence of the environmental conditions. Moreover, I believe that most of the NPF event are strongly influenced by the dynamic of the atmosphere (BL height). Within the FT, the BLH is dragging enough vapors to condense while in an urban site that would disperse a bit the preexisting particles. I think this is important to discuss that in the manuscript. For exemple, How often are IZO and SNS within the BL ? Are all the events occurring within the BL or just when the BLH is reaching the station ?

**We agree with the reviewer that the studied sites are extremely different and present very different environmental conditions, pre-existing aerosol particles and precursor vapours and, therefore, different formation mechanisms are expected at each site. In this sense, environmental conditions are factors that, together with the concentrations of condensable vapours, will determine NPF events at each site. In fact, P10 L8 specify that environmental conditions would, at least partially, explain the observed differences between the studied sites in the NPF characteristics (frequency, formation rate, growth rate, etc):** "*In addition to the aforementioned local differences in the emissions of precursor vapours and in the environmental conditions that could favour or inhibit the formation of new particles and explain the differences in the NPF occurrence frequency between the studied sites, long-range air masses transport has been identified to play different roles on the occurrence of NPF events depending on the study area*".

**All the sites presented here have previously reported studies on the occurrence of NPF and the factors that might promote/inhibit NPF events at each specific site (IZO: García et al., 2014; SNS: Casquero-Vera et al., 2020; MSY: Carnerero et al., 2019; AMM: Hussein et al., 2020; HAS: Hakala et al., 2019). According to these studies, some environmental variables have opposite impacts on NPF events. For example, García et al. (2014) showed that the occurrence of NPF events at IZO is associated with low RH conditions. Similarly, at HAS, Hakala et al. (2019) showed that the occurrence of NPF events coincide with the minimum on the daily variation of RH but the formation rate increase with increasing RH. In contrast, Hussein et al. (2020) showed that the highest NPF event frequency coincides with the maximum in RH values. In this sense, as pointed out by the extensive review of Kerminen et al. (2018), contrasting effects of environmental conditions have been observed worldwide, and this feature is likely due to the fact that environmental conditions would cause simultaneous environmental-conditions-dependent processes that may either enhance or suppress NPF process. For example, an increase of temperature would increase 1) biogenic emissions of aerosol precursor vapours into the atmosphere and their oxidation to low-volatility vapours or 2) the diurnal development of the BL, both processes producing opposite effects on NPF events (e.g. Kerminen et al., 2018). In fact, the situation is likely very complex, since in many locations also air masses originating from very different source areas tend to be characterized by different environmental conditions and, therefore, NPF events show very different responses between different studies. Therefore, we think that the inclusion of this analysis is more suitable for studies focusing**

on a single site and that the inclusion of this analysis will greatly lengthen the work and will not provide additional insight in the role of desert dust on NPF occurrence, which is the focus of this study.

As pointed by the reviewer and previously discussed, BL would have different effects on the occurrence of NPF events at each specific site. Particularly, at mountain sites the BL is not well defined and even if the BL height remains lower than the mountainous ridges, thermally driven winds develop along slopes, or in valleys or basins and these winds are able to bring BL air masses up to mountainous ridges and summits (Collaud Coen et al., 2018). In this sense, previous works (García et al., 2014; Casquero-Vera et al., 2020) have shown that the occurrence of NPF events at two high altitude sites studied here (IZO and SNS sites) are associated with the transport of precursors and pre-existing particles from lower altitudes during daytime. This was stated in the old version of the manuscript in P8 L20 ( "*Aerosol particles and vapours observed at IZO station are mainly emitted at lower altitudes by natural and anthropogenic sources and then transported to the station by orographic thermal-buoyant upward flows during daytime (García et al., 2014)*") and P10 L1 ("*The high NPF event frequency observed at SNS during summer (69%) is associated with the high intensity of solar radiation and the increase of precursor gases transport from lower altitudes during summer season (Casquero-Vera et al., 2020; 2021)*"). However, the study of the effect of the BL or transport from lower altitudes requires a specific study at each site that would require additional instrumentation and measurements to address the effect of the BL/transport and identify where NPF events ocurrs (inside or above the BL) at each specific site.

In light of the above, we consider that constraining data filtering/screening would affect at each site in opposite ways. Furthermore, including sites with extremely different conditions enhances the results of this study, showing that the occurrence of NPF events is highly frequent during desert dust outbreaks at urban, remote and high-altitude sites. However, as we pointed out in P22 L3-6, the concentration of precursor vapours is a key factor that would allow us to investigate the mechanisms at each site during dusty and non-dusty conditions and would let us to understand the chemical and physical processes occurring at each specific site and further experiments with 1) state-of-the-art gaseous, 2) particulate physical and chemical properties measurements and 3) environmental factors are still needed.

3) I have issued with the methodology used here. First, the authors stated that the Dust event are characterized using simulation results and confirmed with OPS/APS data from each station (P7 L20). I think it would be valuable to show the N200 or the PM10/PM1 during non-dusty, dusty and intense dusty days. Indeed, the dust SD could be different from event to event but I guess most of the dust diameter would be, as you suggested, larger than 200nm. Does the Dust event remains the whole day for all cases?

We include this information in the new version of the manuscript, but due to the lack of $PM_1$ and $PM_{10}$ mass concentration measurements at the different sites, instead of $PM_{10}/PM_1$ as suggested by the reviewer, we included the box and whisker plot of total condensations sink (CS), coarse condensation sink ($CS_C$) and fine condensation sink ($CS_F$) recorded at each

measurement site during non-dusty and dusty conditions days. As it is well known, dust intrusions produce a large increase in the coarse mode particle number concentration (particles with diameter >1 μm) and much more pronounced increase in the surface concentration (that is related to the condensation sink) and volume concentration (mass concentration) of coarse mode particles, leading to a significant increase in $CS_C$ and $PM_{10-1}$. So, for detecting the presence of dust at each site, in addition to satellite images and models, we confirmed the presence of dust at ground level with 1) the increase of coarse mode particles (increase of $CS_C$) and 2) the increase of the ratio between the coarse and total condensation sinks ($CS_C/CS$ ratio). Following the reviewer suggestions, in the supplementary material we included Figure R1 (as Fig. S3 in new version of supplementary material), showing as expected the pronounced increase of $CS_C$ during dusty condition at all studied sites, and we included in P8 L3-8: *"To this end, ground level measurements of fine and coarse aerosol size distributions by SMPS/MPSS and APS/OPS are used. As it is well known, dust intrusions produce a large increase in the coarse mode particle number concentration (particles with diameter >1 μm) and much more pronounced increase in the surface (related to the condensation sink) and volume concentration (mass concentration) of coarse mode particles, leading to a significant increase in $CS_C/CS$ and $PM_{10-1}/PM_{10}$ ratios. In this sense, desert dust intrusions are confirmed at each site if a significant increase of 1) the coarse mode condensation sink and 2) the $CS_C/CS$ ratio is observed."*

And in **P13 L12-13:** **"***In fact, the CSC obtained during dusty conditions was on average 21, 4.5, 2.5, 1.5 and 2 times larger than that observed during non-dusty conditions at IZO, SNS, MSY, AMM and HAS, respectively (Fig. S3).***"**

[Figure]

*Figure R1. Box and whisker plot of CS, $CS_C$ and $CS_F$ recorded at each measurement site during non-dusty and dusty conditions days. The line represents the median of the data and the lower and upper edges of the box represent the 25th and 75th percentiles of the data, respectively. The length of the whiskers represents the 1.5× interquartile range, which includes 99.3 % of the data.*

**On the other hand, since our interest is to study the impact of desert dust events on NPF characteristics, we compared the NPF characteristics observed during non-dusty and dusty conditions at each measurement site. For these comparisons we used NPF events whose entire process occurs in the presence of desert dust (from the beginning to the end of the NPF process). In this sense, dust influence doesn't need to remain the whole day at the measurement site. To clarify this point, in P11 L1-4 we added:** *"To analyse the potential impact of desert dust intrusions on the occurrence of NPF events, Figure 2 shows the NPF event frequency for the whole measurement period and the percentage of NPF event days observed during non-dusty and dusty conditions at each measurement site.* *It is worth noting that NPF events are considered as NPF events occurring during dust events only if the whole NPF process occurs in the presence of desert dust."*

**Finally, we think that the metric N200 proposed by the reviewer is not adequate for identifying the presence of dust since this metric can include a significant contribution from particles of other origin than dust (see comment #28 for more information about this).**

Moreover, as it is not clearly stated, I believe that the authors selected all the values observed during event and non-event days as well as during non-dusty, dusty and intense dusty days and compare it as is. However, all those NPF events can have a strong impact on CS, N50 or N200. If you want to use the CS as a limiting parameter for NPF occurrence, could you please compare the CS before the NPF event start? It's usually called the CS2 (2 hours prior to the event) and it would be much more interesting than using tall the daily recorded values CS. Indeed, the SD clearly show that the particles are not growing to the same size. Therefore, some newly formed particles will have a clear effect on the CS (diameter reaching 50 or 60 nm) while the newly formed particles that remained below 30nm won't have the same effect. That would lead you to a biased interpretation. Same advices for N50 : You are trying to better understand the impact of those NPF events on the N50. Then to do so I would compare N50 before the event starts and after the event starts. Indeed, the N50 seems to be similar during NPF Event/NonEVENT/Dusty/nonDusty. As presented right now it's hard to understand if the N50 differences are due to the NPF events or the dust events.

**We apologize for the lack of detail in the analysis of the CS. We agree with the reviewer that if this time window is not applied, the growing particles would impact on the comparisons of the CS. For this reason our analysis was done as the reviewer commented, averaging the CS for the three hours window prior to when the events tend to occur at each specific site. To clarify this point, we added the following sentence (P12 L19-20):**

"*In order to investigate the role of pre-existing aerosols on the process of NPF, CS has been retrieved for each PNSD and averaged each day for the 3 hours before the start time of NPF events at each measurement site.*"

**Similarly, the N50 was averaged in the time window at which particle growth from NPF events taper off at each site as it was specified in P18 L19-21:**

"*In order to estimate CCN concentrations at the different studied sites, median particle number concentrations of particles larger than 50 nm in diameter were retrieved for each day at each measurement site in the 15:00-18:00 h time window (N50 from now on; time window at which particle growth from NPF events taper off; Fig. S5).*"

In addition, we acknowledge the reviewers' suggestion to compare N50 before and after the NPF events (denoted here as N50_prior and N50_after, respectively) in order to better estimate the influence of NPF events on potential CCN budget. In fact, we already considered this method for separating the contributions of NPF and desert dust to CCN during NPF events occurring during dust periods. Other authors also used this method to estimate NPF contribution to CCN comparing the CCN concentrations before and after the NPF event, and reported several issues on its application (e.g., Lihavainen et al., 2003; Laakso et al., 2013; Peng et al., 2014; Ren et al., 2021; Rejano et al., 2021). The main limitations of this approach arise from the fact that there are other factors contributing to the CCN population simultaneously, which leads inaccurate CCN estimation. Specially, we observed two main limitations/changes that do not allow us to apply this method at our measurement sites: 1) changes in boundary layer height and/or 2) changes in primary emissions.

First, this method considers that there are no changes in boundary layer height. However, NPF events coincide with the period when the PBL height changes greatly, leading to significant change in preexisting aerosol load, which will result in biases in the CCN enhancement factor, N50_after/N50_before (e.g., Ren et al., 2021). In fact, the preexisting aerosol loads at high mountain sites (as in SNS and IZO) show a marked diurnal pattern due the atmospheric vertical transport from lower altitude, which would result in a significant overestimation of CCN enhancement factor at these sites.

In addition, this method considers that there are no changes in primary emissions. However, this assumption is not completely valid in the studied sites. For example, when applying this method to AMM (suburban site), we observed that the enhancement factor (N50_after/N50_before) is frequently below 1, specially during the cold period (Fig. R2). This result is observed for NPF events and non-event days and suggests a significant impact of primary emissions before the NPF event starts at this location. In fact, Fig. S5 shows a large increase in particle number concentration before the NPF event starts, suggesting that these primary particles would have an impact on the estimation of the enhancement factor. Therefore, unreasonable enhancement factor (N50_after/N50_before) results were obtained using this method at all the studied sites.

Although we could not estimate the individual contributions of NPF and dust to CCN, the results of this study clearly reveal that 1) NPF events contribute significantly to the CCN budget during both dusty and non-dusty conditions and 2) the contribution of NPF events to CCN budget is larger during dusty conditions than during non-dusty conditions. Actually, it is hard to separate the contributions of NPF to CCN (especially during desert dust events) and this question is one of the remining open questions that need of further investigation. For this reason, in P21 L20-23 we added the sentence: "*While this study reveal a significant impact of class I NPF events on CCN budget during desert dust outbreaks at the studied sites, several questions remain unexplained, such as 1) how desert dust influences cloud formation?, and 2) what is the separate contribution of desert dust and NPF to the CCN budget during NPF occurring during desert dust periods?.*"

[Figure]

**Figure R2. Time series of the enhancement factor at AMM. Red line as reference of enhancement factor = 1.**

4) As it is not stated clearly, I believe that the CS was calculated assuming that the condensable vapors have molecular properties similar to sulfuric acid and therefore this parameter is not "only" depending on the aerosol size distribution as stated by the authors.

**Effectively, the CS was calculated assuming that the condensable vapours have molecular properties similar to sulfuric acid as it was stated in P7 L16-18:** "*...where D is the diffusion coefficient of condensable vapour, that is assumed to be sulfuric acid, and $\beta_M$ is the transitional correction factor (Fuchs and Sutugin, 1971) which is dependent on the mean free path of vapour molecules and aerosol diameter...*".

**In this sense, we agree with the reviewer that the CS depends on the particle and vapours characteristics. However, we used the term "only" to refer to the *particles'* properties and to state that the morphology, physical state, hygroscopicity or chemical composition of aerosol particles effects are, traditionally, not considered in the calculation of CS. Thus, we have clarified the sentences where it can confuse the reader and removed the term "only" where it can be misunderstood, e.g.:**

**P3 L4-6:** "*However, traditionally, the effects of the morphology, physical state and chemical composition of the pre-existing particles are ignored in the calculation of CS.*"

**P12 L14:** "*In general, the CS is usually calculated by considering  the PNSD measured by MPSS*"

**Response to referee #2 minor comments:**

5) P3 L5 : "only the particle number size distribution is considered." Again It's not entirely true…

**Corrected, see our previous comment.**

6) P8 L 19 "the highest frequency was observed at HAS." Are you not discussing SNS results in purpose ?

**In P8 L19 we discussed the annual NPF frequency. Thus, SNS is not discussed here because the data does not cover a whole year and, therefore, it is not *annual* NPF event frequency. In this sense, SNS results were discussed in P9 L16. However, we have clarified this as follows (P9 L3):** "*The highest annual NPF event frequency (68%) was observed at HAS*

5 *(SNS 69% but does not cover a whole year), twice that at AMM...*"

7) P9 L11_15 : I found it weird to read (50% of the paragraph) about seasonal variability within a paragraph where you state it has no seasonal variability…

**Ok. We rewrite this paragraph as follows (P9 L15-24):**

10 *"The NPF frequency shows a clear seasonal pattern at AMM, IZO and MSY sites, with NPF events occurring mainly during late spring and early summer. This seasonal variability is in agreement with the global review of NPF events by Nieminen et al. (2018) who reported the highest NPF occurrences during spring/summer and the lowest ones during autumn/winter. Several previous studies related such seasonal variability in the NPF event frequency to the seasonal variabilities in solar radiation and emissions of precursor gases from biogenic sources (e.g., Dada et al., 2017; Jokinen et al., 2022). In contrast,*

15 *the frequency of NPF events at the HAS site do not show this clear seasonal pattern, being relatively high throughout the year (53-77%; Fig. S1 in the Supplement). The prevalence of clear skies and high solar radiation throughout the year in combination with a persistent transport of anthropogenic precursor vapours from neighbouring coastal urban and industrial areas may explain the absence of an evident seasonal pattern of NPF frequency at the HAS station (Hakala et al., 2019)."*

20 8) P10 L 14/15 : Not well said : "The results agree with the periods "

**Thanks, we changed by (P10 L31-32): "***These periods coincide with the highest frequencies of* desert *dust outbreaks observed at AMM and HAS* (Fig. S2).**"**

9) P10 L21 : Why not showing the CSc and CSf ?

25 **As mentioned in comment #3, we have added in the supplementary material the figure with CS, CSc and CSf for dusty and non-dusty conditions at each site.**

10) P11 L 5-6 : "Therefore, the occurrence of NPF events at these sites is probably not only limited to highly polluted dust plumes (as suggested by Nie et al., 2014), and they can even occur in remote sites during desert dust outbreaks." As stated just above this is true only if there is enough vapours to condense !

**Yes, the NPF process occurs only if there are enough condensable vapours. However, we refer here to the fact that NPF events would not need *highly* polluted dust plumes as suggested by Nie et al. (2014), and even at remote sites with a low (but sufficient) amount of vapours would be enough for the occurrence of NPF events. In P11 L22-24 we have clarified this point as follows:** "*Therefore, NPF events can even occur frequently in remote and high-altitude sites during desert dust outbreaks and, despite sufficient amount of precursors vapours is necessary, the occurrence of NPF events is probably not only limited to highly polluted dust plumes as suggested by Nie et al. (2014).*"

11) P11 L9-10 : "Thus, this result evidences that the intensity of desert dust can limit the occurrence of NPF events at IZO and AMM, which could be explained by a significant reduction or a limited amount of precursor gases to compete with the increase of available surface of pre-existing particles. " So your hypothesis is that there are no condensable vapors within the dusty air mass ? IZO is within the FT or the BLH at that time ? At AMM, urban site, a lot of condensable vapors are available and when the dusty air masses are coming that will increase the number of preexisting particles and would block the process? I found it difficult to state that in one sentence without proof. That would be interesting to focus on those days to better understand the role of the environmental parameters known to play a large role such as RH, radiation etc…

**As the reviewer pointed, "*a significant reduction or limited amount of precursor gases during intense desert dust events that cannot compete with the increased amount of pre-existing particles*" is an hypothesis and this cannot be proven with the instrumentation available at the measurement sites. In this sense, we have rewritten the sentence as follow to clarify that is just an hypothesis (P11 L26-30):**

*"Thus, this result evidences that the intensity of desert dust can limit the occurrence of NPF events at IZO and AMM, which may be explained by a significant reduction or a limited amount of precursor gases to compete with the increase of available surface of pre-existing particles. However, additional data (e.g., precursor gas vapours measurements, aerosol mineral and chemical composition, meteorological information, etc.) are needed to confirm this hypothesis."*

**Regarding if IZO is within the FT or the BLH, García et al. (2012) showed that during daytime IZO station is affected by the transport of precursor gases from lower altitudes and NPF events are triggered by this transport from the boundary layer, so we don't consider IZO is within the FT during daytime.**

**About the environmental parameters, we refer to our response of reviwer's comment #2. As previously discussed, the role of environmental parameters is site dependent and we consider that this analysis without further information and applying process-scale modelling approaches would not provide clearer insights on the mechanisms promoting the**

**occurrence of NPF events at each specific site. Therefore, we consider that further studies at each site with additional data of meteorological variables, gaseous and particulate physical and chemical properties measurements is still needed to better understand the role of desert dust on the NPF mechanisms during dusty conditions.**

12) Figure 2 :Can you change the way you plotted it so it could help the reader? Please add some texture for event, Non event and undefined days ? The colors for non Dusty/dusty and Intense dust are really close. Can you pick another color ?

**We have changed the colours by using colour blind friendly palette and also added texture to undefined days.**

13) P12 L1-3 : You have said that multiple times and this is I think not necessary..

**We have removed these lines.**

14) P12 L25 and P12 L30 one more repetition. I believe that this is not 21 times more in IZO but just 2 times more ! Again here I would use the CS2 so you can actually use it as a parameter that could prevent or not the NPF event.

**The coarse condensation sink ($CS_C$) is 21 times larger during dusty days at IZO (P13 L12) and total condensation sink (CS) is 2 times larger at this site (P13 L18). Again, we used the CS3 (three hours time window; please see our response regarding CS2 in comment #3).**

15) P12 L31 : You can't say that the CS is increasing just by the removal of fine particle through coagulation without proving it. Could you plot the CS as a function of the diameter at different moment of the day (before /during and after the NPF event). From here you could state this strong conclusion supported by results : "Thus, these results highlight the impact of desert dust outbreaks into the CS and the importance of considering coarse mode particles for adequate CS calculations at desert dust-influenced areas."

**We agree that we cannot ensure that the reduction of $CS_F$ is due to the increase on the coagulation ratio. In fact, an increase of the coagulation ratio can only be observed under controlled conditions (as chamber studies). Therefore, to avoid speculation, we removed this statement.**

**The conclusion that desert dust impacts into the total CS has been previously discussed and we have added the differences of the $CS_C$ and $CS_F$ during dusty and non-dusty conditions (Fig. R1), supporting that $CS_C$ contribute significantly to the total CS at dust-influenced areas during dusty conditions.**

16) P13 : Here I think that all the explanations are dubuious since you used all CS values recorded during the day. I would strongly recommend to recalculate the CS 2 hours prior to the NPF events.

**We apologize for the misunderstanding and, as stated in our response to comment #3, the CS was retrieved in 3 hours prior to the NPF events.**

17) P14 L12-14 : This should probably go within the introduction such as the sentence from P14 L19-21. I would appreciate having more information about the TiO2 impact.

**Following reviewer's suggestion, we have moved this paragraph to the introduction section and we added some information about the TiO2 impact. Thus, the introduction has been modified as follows (P3 L17-29):**

"*The climatic effects of desert dust and atmospheric NPF have been thought to be disconnected from each other, however, high dust loadings can affect NPF in opposing ways. Desert dust can inhibit the NPF process 1) by reducing UV radiation and thus reducing photochemical processes and 2) by increasing the scavenging rates of precursor vapours by desert dust particles and thereby supressing or reducing the new particle formation and growth rates during NPF events (de Reus et al., 2000; Ndour et al., 2009). However, desert dust can also enhance the NPF process by enhancing the formation of OH and other radicals by the presence of catalyst components in the desert dust particles, favouring oxidation reactions and therefore promoting the occurrence of NPF events during dusty conditions. In fact, several authors revealed that $TiO_2$ and $Fe_2O$ (which are common components of desert dust) act as photocatalysts and under UV light could promote the heterogeneous oxidation of $SO_2$ and the subsequent formation of gaseous $H_2SO_4$, inducing NPF (e.g. Dupart et al., 2012 and references therein). These components, acting as catalysts, are not consumed in the photocatalytic reaction and can accelerate atmospheric photochemistry repeatedly. Furthermore, more recent laboratory study showed that the presence of $TiO_2$ greatly promotes NPF and can enhance the particle formation and growth rates by a factor up to 3 and 2, respectively (Zhang et al., 2023). However, clear association between desert dust loadings and the occurrence or strength of NPF has not yet been established.*"

**and in P14 L23-P15 L3:**

"*In addition to influencing the frequency of occurrence of NPF events, desert dust could change the strength and other characteristics of the NPF events occurring during dusty conditions. The strength of a NPF event is determined by its duration, the rate at which new particles are formed (J) and the growth rate (GR) of these particles into larger sizes. In this sense, the increased contribution of desert dust particles during dusty conditions could imply the inhibition or enhancement of the NPF process. Thus, in the following, the possible impact of desert dust on the characteristics and strength of NPF event occurred in the different studied sites was analysed.*"

**More information about the TiO₂ impact is also given in our response to the following comment.**

18) "In fact, recent laboratory study (Zhang et al., 2023) revealed that TiO2 contributes significantly to the formation of gaseous H2SO4 by increasing the GR and J by up to a20 factor of 2 and 3, respectively, in the presence of TiO2." TiO2 is in the particular phase. How could it influence the production of H2SO4 ? You mean that there are more H2SO4 because the GR increase ? So there is a particular chemistry that would enhance the condensation rate of some vapors (which ones ??? ) and in the same time there will be more H2SO4 within the vapor phase ?

**Sorry, this sentence does not clearly express what we meant. Several authors revealed that TiO₂, which is one of the common components of mineral dust, can act as photocatalyst promoting the heterogeneous oxidation of SO₂ and the subsequent gaseous H₂SO₄ formation which induce NPF (e.g. Dupart et al., 2012 and references therein). Furthermore, more recent study showed that the presence of TiO₂ greatly promotes NPF and enhances particle formation rate (J) and growth rate by a factor of 2 and 3, respectively (Zhang et al., 2023). So, to be clearer in P3 L24-28 we changed by**

*"In fact, several authors revealed that TiO₂ and Fe₂O (which are common components of desert dust) act as photocatalysts and under UV light could promote the heterogeneous oxidation of SO₂ and the subsequent formation of gaseous H₂SO₄, inducing NPF (e.g. Dupart et al., 2012 and references therein). These components, acting as catalysts, are not consumed in the photocatalytic reaction and can accelerate atmospheric photochemistry repeatedly. Furthermore, more recent laboratory study showed that the presence of TiO₂ greatly promotes NPF and can enhance the particle formation and growth rates by a factor up to 3 and 2, respectively (Zhang et al., 2023)."*

[Figure]

**Figure R3. Scheme of reaction mechanism presented on Dupart et al. (2012).**

19) P15 – L13-15 : That was expected to have stronger values of CS during dusty days. Again, I would present here CS2 so you do not take into account the NPF event influence on the GR …

**We again apologize for the misunderstanding. As pointed before, our results present the CS averaged in the three hours window when the events tend to occur at each specific site (please see our response to comment #3 about CS).**

20) Figure 4 : At MSY you have only one event during intense dust days. However, from figure 2 you have at least 10 NPF events. Please help us to understand… From what I understood the whole period here corresponds to events during non dusty right ? It can't be the dusty and non dusty NPF events otherwise the number would be much larger ! Change the legend to state it clearly what it referes to … . But then if this is events during non dusty days it cshould correspond to the numbers on Figure 3 and 7 which is not…

**As stated in Fig. 4 caption, the GR and CS of this figure refers only to Class I events (those when GR and J could be calculated). In contrast, Fig. 2 shows "all" the events. In addition, P15 L15-17 we stated that the previous analysis of CS (Fig. 3)** "*was done for the whole dataset, whereas GR was only retrieved for NPF class I events. Thus, in order to identify the possible role of CS on the growth rates, the CS has been retrieved for the corresponding NPF class I events during non-dusty, dusty and intense dusty conditions (Fig. 4b)*".

**To show the coherence between the different figures, we can for example analyse the results corresponding to IZO. In this sense, Figure 4 shows that there is 43 class I events during the whole period,** *30 during non-dusty conditions* **and** *13 during dusty conditions* **at IZO (3 of these 13 cases correspond to intense dusty conditions). If we compare with Fig. 7, we can see that there are the same numbers for class I category during dusty (13 days) and non-dusty (30 days) conditions.**

21) P18 L9. A SS of 0.75% is an extremely large value and correspond to very very fast-growing clouds. I'm not sure these SS are observed in reality. That means that you choose an activation diameter that is really low. So ou are probably overestimatin the number of CCN available with this threshold. It needs to be stated somewhere.

**While typical SS values in clouds are below 0.75%, several studies have shown SS > 0.75% (even above 1%), not necessarily meaning very fast-growing clouds (e.g. Hammer et al., 2014; Hudson and Noble, 2014). However, the critical diameter of 50 nm is the lower-range of critical diameters (e.g. Kerminen et al., 2012) and particles smaller than 50 nm can only be activated at high water vapor saturation ratios. Thus, we have added the following sentence in P18 L15-18 in order to clarify that we are referring to the upper limit of CCN concentrations:**

"*In fact, Rejano et al. (2023) observed critical diameters of 50-60 nm at SS=0.75% for different aerosol sources and, therefore, $N_{50}$ described below was considered a good CCN proxy at high SS values and particles smaller than 50 nm would only be activated at exceptionally high water vapor supersaturation ratios. Thus, the estimated CCN concentration at SS=0.75% represents the upper limit of the estimated CCN.*"

22) Section 3.5 again here I would plot the N50 before and after the event start to clearly separate the impact of the newly formed particles on the N50. So you could estimate the impact of NPF during dusty days in comparison to non-dusty days.

**As discussed before, two main limitations/changes do not allow us to apply this method at our measurement sites: 1) changes on primary emissions and/or 2) changes of boundary layer height. For more detail, please see our response to the comment #3.**

23) P19: Do you know why the newly formed particles do not grow to larger size ? It seems from Figure S4 that there is a threshold different from site to site for the diameter reached by the end of the NPF events.

**There are two possible explanations for the maximum size that the NPF events reach at each location. On the one hand, it could be that the production of precursor vapours stops at a certain hour and, therefore, particles do not continue growing above that certain size. On the other hand, it could be that the air mass where NPF events occurs is not observed at the measurement site after a certain time. However, with fixed measurement sites we are not able to know which of the cases we are observing. Therefore, we always refer to the effect that dust have on the CCN budget, specifically, at our measurement sites, because we are not able to know if those particles will continue growing somewhere else or if the process has stopped.**

24) Figure S4 is an average over dusty and non-dusty days right ? It seems that SNS show more large particles (around 100nm) during non dusty days. Why is that ? I don't see a clear difference for AMM. Can you comment ? IZO the SD does not show a banana shape. So I'm not sure it could be sorted as class I event. Can you comment on that ?

**Yes, Fig. S4 (Fig. S5 in the new version) shows the comparison of daily average pattern of NPF that occurred during dusty and non-dusty conditions.**

**As pointed by the reviewer, the Fig. S5 shows the presence of particles with sizes around 100nm at SNS during both dusty and non-dusty days and it seems that the concentration of these particles is larger in non-dusty days. It also shows the presence of these particles throughout the day, both on dusty and non-dusty days, which may indicate that these particles are background particles. Change in the dilution effects and the possible increase of the scavenging of pre-existing particles by dust particles during dusty conditions may explain the difference in the concentrations of these particles between dusty and non-dusty days.**

**The differences in AMM are subtle as it has been previously shown in the results of GR, J and CS during dusty and non-dusty conditions, probably due to the less influence of dust on NPF characteristic at this site.**

**Finally, IZO shows a very smooth "banana" shape due to the low GR observed at this site, in agreement with previous results at this site (Garcia et al., 2014).**

25) Figure 6 : I would not show this figure but I would add a table with those numbers. Maybe you can insert it in Table 1.

**Thanks for the suggestion but we consider that this figure is more intuitive, using similar presentation than Fig. 2. In any case, we think that including this information in Table 1 will hinder a smooth reading of the paper, since the interpretation and discussion of this information (on page 19) would be very far from table 1 (on page 9).**

26) P20 and Figure 7 : According to Figure7 there are no statistical differences between the Non dusty and dusty days for N50 at IZO, AMM and HAS. Indeed, the boxplots show similar 0.25 and 0.75 percentile behaviors for these 3 sites. So this is hard to draw strong conclusions on the effect of dusty events on the NPF efficiency to increase N50.

**If we understood correctly, the reviewer refers to the differences observed during class I NPF events for non-dusty and dusty days. If so, the reviewer is right and the differences in N50 between class I NPF events during dusty and non-dusty conditions are not statistically significant at IZO, SNS, AMM and HAS (as indicated by Mann-Whitney test at 0.01 significance level). This is due to the substantial difference in the background aerosol conditions between dusty and non-dusty conditions which mask the effect of NPF during dust events on N50 and does not allow us to better elucidate the impact of the NPF during dusty conditions on N50. For this reason, the comparison of N50 during class I events between dusty and non-dusty conditions was not discussed and not used for the conclusions drawn in this section. Therefore, for investigating the impact of NPF during dusty conditions, we focused our discussion on the comparison of N50 for class I NPF days and non-event days during dusty conditions, and we do the same analysis for non-dusty conditions, in order to have more similar background aerosol conditions.**

27) Again add texture instead of just having a light blue and blue colors…

**Ok. We have changed the colours by using colour blind friendly palette.**

28) The N50,dusty is sometimes lower than N50,non-dusty (SNS, AMM and HAS ). So first I'm not sure I'm able to understand that especially since you stated that you selected the dust evet by using the SSD of the coarse mode. Now, supposedly you have a dust event for the whole day (it's not stated what is the duration of these dust events). So the N200 should be higher during dusty events (need to show that to draw the later conclusions). How is the N50-200, non-dusty in comparison to N50-

200, dusty ? Again as some newly formed particles grow larger than 50 nm I can't tell if this is due to the increase is solely coming from the NPF event during non-dusty days that could lead to larger particles due to more vapors available to grow. I strongly advice to compare the increase of N50 before in comparison to after the NPF events and find a way to normalized it according to the dust concentration so we can clearly understand the effect of NPF/Dust on the CCN concentration.

**Effectively, the N50_dusty is sometimes lower than N50_non-dusty (SNS, AMM and HAS) and, as we commented before (comment #26), this is because the background aerosol number concentration of particle with diameters >50 nm during non-dusty days is higher than during dusty days. For this reason, the comparison of N50 between dusty and non-dusty conditions was not discussed and not used for the conclusions drawn in this section. In this sense, for investigating the impact of NPF during dusty conditions, we focused our discussion on the comparison of N50 for class I NPF days and non-event days during dusty conditions in order to have more similar background aerosol conditions.**

**On the other hand, it is well known that dust intrusions produce a large increase in the coarse mode particle number concentration (particles with diameter >1 μm) and much more pronounced increase in the surface (~condensation sink) and volume concentration of coarse mode particles, leading to a significant increase in $CS_C/CS$ ratio. In this sense, we think that the metric (N200, particle number concentration of particles with D>200 nm) proposed by the referee is not adequate for identifying the presence of dust since this metric can include a significant contribution from particles of other origin than dust. Thus, the N200 metric should not be necessarily higher during dusty events as the reviewer stated. In this sense, for confirming the presence of dust in each site, in addition to satellite images and models, we used $CS_C$ and $CS_C/CS$ ratio (see our response to comment #3).**

**As mentioned before (see our response to comment #3), the method proposed by the reviewer for estimating the contribution of NPF occurring during dust events to the N50 provides unrealistic results (negative N50) due to the significant daily change in the boundary layer height and local emissions. In this sense, it is hard to separate the individual contributions of NPF and dust particles to N50 during NPF occurring during dust episodes and this question is one of the remining open issues that need to be addressed in the future. In any case, the results of this study clearly reveal that NPF events occurring during dust events contribute significantly to CCN. For more detail about this question and the corresponding changes done, please see our response to comment #3.**

**References**

- Collaud Coen, M., Andrews, E., Aliaga, D., Andrade, M., Angelov, H., Bukowiecki, N., Ealo, M., Fialho, P., Flentje, H., Hallar, A. G., Hooda, R., Kalapov, I., Krejci, R., Lin, N.-H., Marinoni, A., Ming, J., Nguyen, N. A., Pandolfi, M., Pont, V., Ries, L., Rodríguez, S., Schauer, G., Sellegri, K., Sharma, S., Sun, J., Tunved, P., Velasquez, P., and Ruffieux, D.:

Identification of topographic features influencing aerosol observations at high altitude stations, Atmos. Chem. Phys., 18, 12289–12313, https://doi.org/10.5194/acp-18-12289-2018, 2018.

- Hammer, E., Bukowiecki, N., Gysel, M., Jurányi, Z., Hoyle, C. R., Vogt, R., Baltensperger, U., and Weingartner, E.: Investigation of the effective peak supersaturation for liquid-phase clouds at the high-alpine site Jungfraujoch, Switzerland (3580 m a.s.l.), Atmos. Chem. Phys., 14, 1123–1139, https://doi.org/10.5194/acp-14-1123-2014, 2014.

- Hudson, J. G. and Noble, S.: CCN and Vertical Velocity Influences on Droplet Concentrations and Supersaturations in Clean and Polluted Stratus Clouds, J. Atmos. Sci., 71, 312–331, https://doi.org/10.1175/JAS-D-13-086.1, 2014.

---

## Referee Report (RR1)

Referee comment on the revised version of "Impact of desert dust on new particle formation events and cloud condensation nuclei budget in dust-influenced areas" by Casquero-Vera et al., https://doi.org/10.5194/egusphere-2023-1238

Casquero-Vera et al. presented an extensive study of the impact of desert dust on the occurrence of new particle formation (NPF) events at five different dust-influenced sites. The authors showed that NPF events occur during dusty and non-dusty conditions, similarly, evidencing that NPF is not limited to highly dust outbreaks. Furthermore, the authors calculated the condensation sink (CS) in both fine ($CS_F$) and coarse ($CS_C$) modes. This calculation showed that the value of CS can be underestimated by 17% if the coarse mode is not considered.

The growth rates ($GR_{10-25nm}$) and nucleation rates ($J_{10nm}$) do not show a clear pattern of the effect of desert dust on the strength of the NPF event occurrence. This suggests that other important parameters might play a role, for example, the concentration of precursor gases, chemical composition, and meteorological parameters. The authors calculated the CCN potential by using the particle number size distribution and found that NPF events contribute to the CCN budget during dusty conditions.

The authors have discussed and carefully attended the comments and suggestions made by the two referees. The quality of the manuscript has improved and therefore should be considered for final publication in EGUsphere.

---

## Author Response (AR2)

**Review of Casquero-Vera et al. "Impact of desert dust on new particle formation events and cloud condensation nuclei budget in dust-influenced areas"**

The authors of this paper would like to thank the reviewer for his/her positive comments and the editor for reviewing the

5   reviewer responses and providing suggestions on how to improve the quality of the manuscript and prepare it for publication.

We have carefully considered the feedback and have made appropriate changes to address remaining technical and minor

changes.